# Future strengthening of the Nordic Seas overturning circulation

Marius Årthun ●[1] ✉, Helene Asbjørnsen ●[1], Léon Chafik ●[2,3], Helen L. Johnson ●[4] & Kjetil Våge[1]

The overturning circulation in the Nordic Seas involves the transformation of warm Atlantic waters into cold, dense overflows. These overflow waters return to the North Atlantic and form the headwaters to the deep limb of the Atlantic meridional overturning circulation (AMOC). The Nordic Seas are thus a key component of the AMOC. However, little is known about the response of the overturning circulation in the Nordic Seas to future climate change. Here we show using global climate models that, in contrast to the North Atlantic, the simulated density-space overturning circulation in the Nordic Seas increases throughout most of the 21st century as a result of enhanced horizontal circulation and a strengthened zonal density gradient. The increased Nordic Seas overturning is furthermore manifested in the overturning circulation in the eastern subpolar North Atlantic. A strengthened Nordic Seas overturning circulation could therefore be a stabilizing factor in the future AMOC.

The Atlantic meridional overturning circulation (AMOC) carries warm and saline waters northwards near the surface and cold, dense waters southwards at depth[1]. The northward branch of the AMOC terminates north of the Greenland-Scotland Ridge that separates the North Atlantic Ocean from the Nordic Seas and Arctic Ocean (Fig. 1). In the Nordic Seas, the warm Atlantic water is gradually transformed by atmospheric heat loss and freshwater input[2–5], leading to the formation of dense overflow waters that form a substantial contribution to the lower limb of the AMOC[6–8]. The Nordic Seas are thus key to the state of the AMOC[9]. Understanding future change in the Nordic Seas is therefore essential, but the impact of anthropogenic climate change on water mass transformation and associated overturning circulation in the Nordic Seas remains little explored.

In the Nordic Seas, there are significant horizontal gradients in water density and the transformation (overturning) of less-dense waters to more-dense waters is best quantified by the meridional overturning stream function in density space[10–12] (Methods). Recent results – from both observations and models – have demonstrated that the cyclonic horizontal circulation that flows across sloping isopycnals dominates the density-space overturning in the subpolar North Atlantic and Nordic Seas[12–15]. These results thus challenge the

importance of open-ocean deep convection in setting the strength of the AMOC. Previous modeling efforts have, however, identified the northward shift of deep-convection sites as a possible source of future overturning changes in the Nordic Seas and Arctic Ocean[16–18].

Here we show that, in contrast to the overturning circulation in the North Atlantic[19,20], the projected density-space overturning circulation in the Nordic Seas in CMIP5 and CMIP6 models shows no persistent decline in the future, and is rather characterized by an increase between 2040 and 2100 as a result of a strengthened horizontal circulation and zonal density gradient. The Nordic Seas overturning circulation changes in synchrony with the Atlantic Multidecadal Variability (AMV) index as the associated redistribution of large scale hydrography translates into changes in water mass transformation and horizontal circulation. The strengthened Nordic Seas overturning furthermore has a detectable and significant imprint on the overturning circulation in the eastern subpolar North Atlantic.

## Results

### Future increase in Nordic Seas overturning circulation
Time series of the simulated strength of the Nordic Seas overturning circulation in depth-space ($MOC_z$) and density-space ($MOC_\sigma$) in CESM-

[1]Geophysical Institute, University of Bergen, and Bjerknes Centre for Climate Research, Bergen, Norway. [2]Department of Meteorology and Bolin Centre for Climate Research, Stockholm University, Stockholm, Sweden. [3]National Oceanography Centre, Southampton, UK. [4]Department of Earth Sciences, University of Oxford, Oxford, UK. ✉e-mail: marius.arthun@uib.no

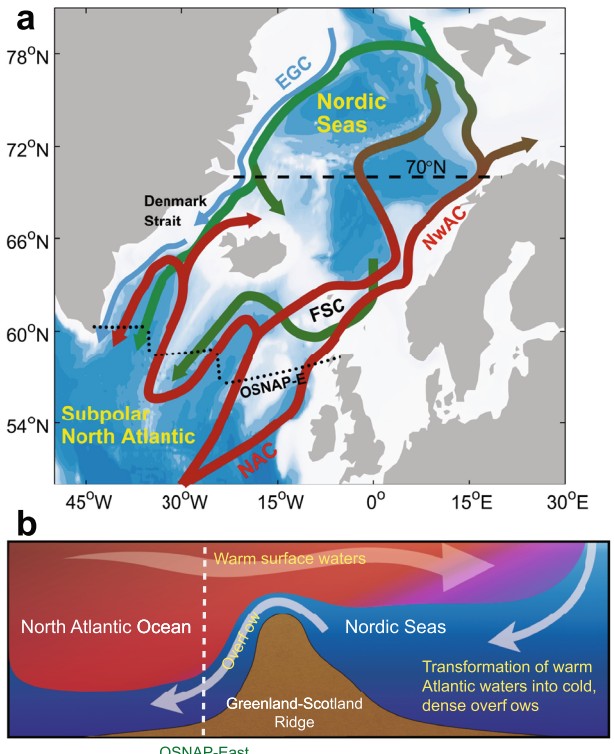

**Fig. 1 | Ocean circulation of the eastern subpolar North Atlantic Ocean and Nordic Seas. a** Schematic of the major ocean currents crossing the Greenland-Scotland Ridge (GSR). The arrows represent the pathways and transformation of warm Atlantic waters (red arrows) in the NAC-NwAC to cold, dense overflows (green arrows) that exit the Nordic Seas through the Denmark Strait and Faroe-Shetland Channel (FSC). The cold, fresh East Greenland Current (EGC) is shown by blue arrows. The sections at 70°N and in the eastern subpolar North Atlantic (mimicking the eastern part of the array from the Overturning in the Subpolar North Atlantic Program[13]; OSNAP-East) used to calculate overturning circulation are shown as dashed black lines. NAC: North Atlantic Current, NwAC: Norwegian Atlantic Current. **b** Schematic representation of the meridional flow across the GSR highlighting how the dense overflow waters from the Nordic Seas contribute to the lower limb of the overturning circulation at OSNAP-East.

LE, quantified at 70°N, are shown in Fig. 2 (corresponding overturning streamfunctions are shown in Supplementary Fig. 1). The present-day density-space overturning in CESM-LE is well simulated compared with ocean reanalyses (Fig. 2; Supplementary Fig. 2). Both overturning time series are characterized by fairly stable values between 1920 and 2000, followed by a subsequent decline. However, whereas $MOC_z$ shows a steady decrease throughout the 21st century, $MOC_\sigma$ shows an increase after the 2040s leading to a recovery of the overturning circulation toward present-day conditions. This increase toward the end of the century is also simulated by CMIP6 models, although we acknowledge the limited number of models included in this estimate (Supplementary Table 1). The density of the maximum overturning circulation gradually decreases as the whole water column at 70°N becomes fresher in the future. There is also a corresponding shoaling of the maximum depth-space overturning circulation. We note that extensive water mass transformation in the Nordic Seas occurs between the Greenland-Scotland Ridge and 70°N, e.g., in the Lofoten Basin[4]. However, we obtain similar overturning characteristics in CESM-LE for a section on the native model grid that cuts across the southern Nordic Seas (Supplementary Fig. 3). These future changes in Nordic Seas overturning circulation arise due to future changes in both the horizontal circulation and the zonal density contrast across 70°N. Next, we therefore explore future changes in horizontal circulation and zonal density gradient, and their underlying mechanisms.

## Horizontal circulation

Overturning in density space includes flow traditionally thought of as gyre circulation, and the difference between the density-space and depth-space overturning circulation in the Nordic Seas approximates the contribution of the horizontal (gyre) circulation to $MOC_\sigma$[12]. The temporal development of the Nordic Seas overturning strength between 1920 and 2100 is closely linked to its horizontal circulation contribution (Supplementary Fig. 4). The divergent behaviour of $MOC_z$ and $MOC_\sigma$ in CESM-LE and CMIP6 models furthermore suggests a strengthening of the horizontal circulation across 70°N after the 2040s. Such future changes in the horizontal circulation are corroborated by the trend in the simulated depth-integrated circulation (barotropic streamfunction; Fig. 3a). During the latter half of the 21st century there is a general slow-down of the horizontal circulation in the North Atlantic, manifested by a weakening of the subtropical and subpolar gyres. In contrast, the horizontal circulation within the Nordic Seas is strengthened, that is, a stronger cyclonic circulation. This increase in gyre strength follows a simulated decrease during the first part of the century (Fig. 3b), consistent with that seen for the Nordic Seas overturning ($MOC_\sigma$). Results from CESM-2C and CMIP6 models show that enhanced horizontal circulation in the Nordic Seas is a common feature across different models and forcing scenarios (Fig. 3b, c), although there is still a large inter-model spread in the future trends in horizontal circulation in the Nordic Seas.

To further evaluate the link between the horizontal circulation and the overturning circulation in the Nordic Seas, we calculate all possible 30-year trends between 2006 and 2100 in both variables, for each ensemble member in CESM-LE. We find that a strengthened overturning circulation is clearly related to a more vigorous cyclonic circulation in the Nordic Seas ($r = 0.76$; Fig. 3d). The correlations are robust for different periods, e.g., considering the first and second part of the 21st century, and for the historical simulation (1920–2005). A future strengthening of the simulated horizontal circulation in the Nordic Seas thus acts to increase Nordic Seas overturning in the latter half of the 21st century.

We now investigate why the horizontal circulation in the Nordic Seas strengthens toward 2100 in CESM-LE, immediately noting that the enhanced circulation after the 2040s is not directly related to the inflow from the North Atlantic, which gradually weakens over the 21st century (Supplementary Fig. 5). Changes in ocean gyre circulation can be driven by both the surface wind stress curl and by changes in the ocean density field. The barotropic circulation in the Nordic Seas has previously been reconstructed using Ekman pumping and geostrophic shear from hydrographic forcing[21]. To investigate the contribution from changes in winds and in the density structure to the increased gyre circulation in CESM-LE after the 2040s we thus calculate the vertically integrated meridional transports associated with the density field (potential energy of the water column; Equation (4), Methods) and from the zonal wind stress, and compare these transports with that obtained from the model's simulated velocities. We find that the spatial pattern as well as the absolute values of the transports driven by density changes (Fig. 4b) are similar to the simulated transports (Fig. 4a). The spatial pattern of meridional transport changes in Fig. 4a, b is also consistent with changes in the barotropic streamfunction used to calculate the strength of the horizontal circulation (Fig. 3). The (Ekman) transport changes resulting from wind stress are negligible (Fig. 4c). The wind stress curl over the Nordic Seas also displays no evident weakening or strengthening between 2006 and 2100 (Supplementary Fig. 6). Hence, the major driver of future changes in the horizontal circulation in the Nordic Seas appears to be changes in the density structure. A similar conclusion has been reached for present-day variability of the subpolar gyre[22–24].

What could be the source of future hydrographic changes in the Nordic Seas? It has previously been shown both from observations[25,26] and models[27] that hydrographic variability in the Nordic Seas is

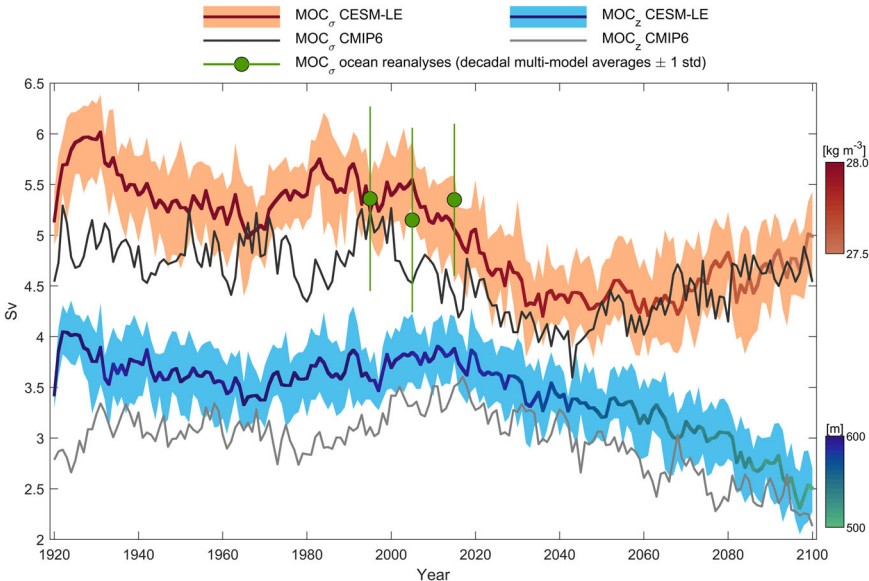

**Fig. 2 | Nordic Seas overturning.** Time series of maximum overturning strength at 70°N in density (MOC$_\sigma$) and depth-space (MOC$_z$) in CESM-LE and CMIP6 models. For CESM-LE, solid lines show the ensemble mean and the shading indicates the interquartile spread. The color of the solid lines indicates the density and depth of the maximum overturning circulation in density-space and depth-space, respectively. Black and grey lines show the multi-model mean MOC$_\sigma$ and MOC$_z$, respectively, from CMIP6 models (SSP585). Green circles show the average density-space overturning for 1993-1999, 2000-2009 and 2010-2019 for the four ocean reanalyses C-GLORSv7, GloSea5, OraS5, and GLORYS2V4. Vertical lines show ± 1 standard deviation across time and reanalyses.

primarily related to changes in the Atlantic Ocean. In support of an Atlantic origin of the temperature (density) anomalies responsible for driving horizontal circulation changes in CESM-LE, the Atlantic Multidecadal Variability (AMV; defined as the second mode of low-frequency variability; Methods) index shows a very similar temporal development between 1920-2100 (Fig. 5a). The historical forced AMV variability in CESM-LE is in close agreement with forced multidecadal AMOC variability in CESM-LE[28] and other CMIP5 models[29], whereas the future development is in agreement with the externally-forced AMV index from the MPI Grand Ensemble using RCP8.5 forcing[30]. The ensemble trend correlation ($r = 0.65$; Fig. 5b) furthermore supports a link between the AMV and horizontal circulation in the Nordic Seas through its impact on density changes. A significant correlation between horizontal circulation changes in the Nordic Seas and the AMV is also found for many of the CMIP6 models (Fig. 5c; multi-model mean correlation of 0.64). The spread in correlation between the CMIP6 models is comparable to the spread between different ensemble members in CESM-LE.

The AMV could also impact ocean circulation in the Nordic Seas through changes in atmospheric circulation, and in particular by driving North Atlantic Oscillation (NAO)-like anomalies[31,32]. However, there are no pronounced future forced changes in the North Atlantic Oscillation and the East Atlantic Pattern in CESM-LE (defined as the two leading principal component time series of winter sea-level pressure anomalies over the Atlantic sector; Supplementary Fig. 6b, c). These results thus suggest that the link between horizontal circulation in the Nordic Seas and the AMV can be understood as multidecadal temperature anomalies in the North Atlantic that are manifested in the surface density field in the Nordic Seas. Further dedicated analyses and experiments would, however, be required to quantify in more detail the relative contribution of wind forcing, convection, and ocean advection to future density changes in the Nordic Seas.

## Horizontal density gradient
The future weakening and subsequent strengthening of the Nordic Seas overturning circulation is consistent with changes in the zonal density gradient at 70°N (200–500 m depth; below the East Greenland Current), showing a weakening (strengthening) gradient before (after) the 2040s (Fig. 6a). The density changes are predominantly driven by temperature changes at the eastern boundary of the Nordic Seas (Supplementary Fig. 7), suggesting that exchanges between the western Nordic Seas and the Arctic Ocean through the Fram Strait (Fig. 1a) only play a minor role in changing the density gradient at 70°N. Considering all 30-year trends between 2006 and 2100, the ensemble trend correlation between MOC$_\sigma$ and the zonal density gradient is 0.91 (Fig. 6b; $r = 0.67$ for 0-500 m). Our results thus demonstrate that the future strengthening of the Nordic Seas overturning circulation is closely related to future changes in the density contrast along 70°N.

## Surface-forced overturning circulation
Surface buoyancy forcing is the primary driver of water mass transformation (overturning) in the Nordic Seas[2,5,9]. To assess to what extent the transformation of surface waters by air-sea heat and freshwater fluxes can explain the future changes in the Nordic Seas overturning circulation, we therefore calculate the surface-forced water mass transformation (MOC$_s$; Equation (3)). This relates the rate of density transformation in a given density class to the surface buoyancy fluxes into that density class over its outcrop area[14,33,34] (Methods). The mean surface-forced water mass transformation in the Nordic Seas (Fig. 7a) closely resembles the time-averaged overturning streamfunction in density space (Supplementary Fig. 1), supporting a clear link between the Nordic Seas overturning and water mass transformation due to surface forcing. Consistent with the temporal development of the overturning circulation in the Nordic Seas (Fig. 2), MOC$_s$ also shows a weakening between the 2010s and 2040s, and a subsequent strengthening toward the 2070s. The decreasing density of maximum water mass transformation is also consistent with the changes in the structure of MOC$_\sigma$. The time series of MOC$_s$ based on 16 CMIP6 models (Supplementary Table 1) also shows decreased water mass transformation toward the 2040s followed by an increase toward the end of the century (Fig. 7d). The magnitude of these changes are furthermore similar for the CMIP6 and CESM-LE ensemble averages. This thus provides additional support for a future increase in the Nordic Seas overturning circulation.

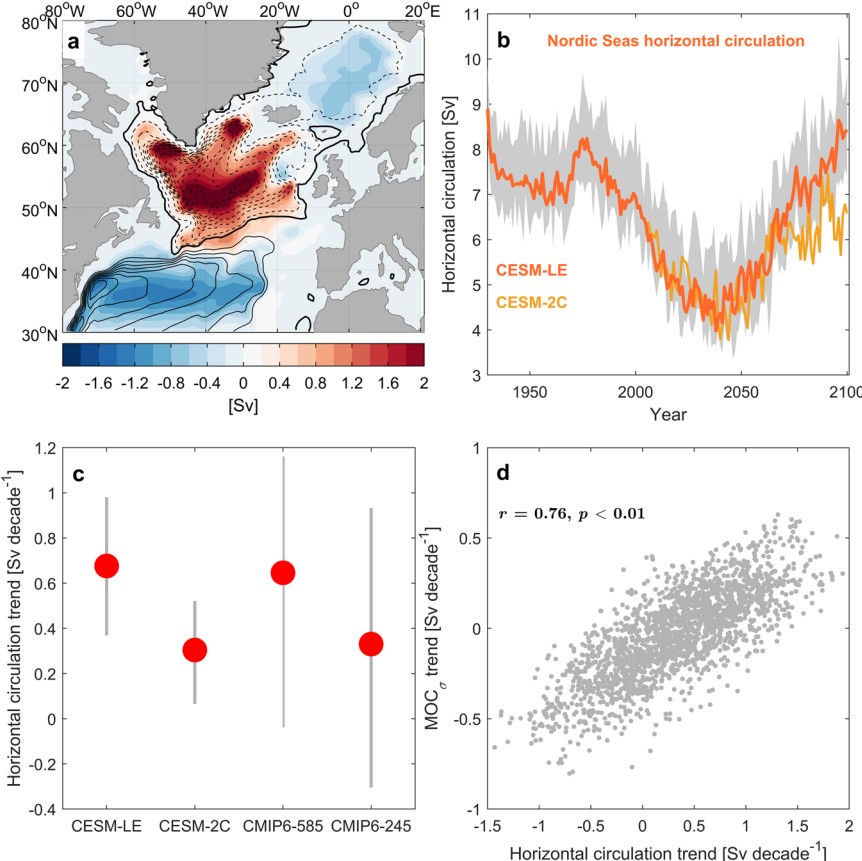

**Fig. 3 | Horizontal circulation. a** Trends in horizontal circulation (barotropic streamfunction; color; unit: Sv decade$^{-1}$) between 2051-2100 in CESM-LE. Mean values for 2006–2100 are shown by contour-lines (contour interval: 5 Sv; dashed lines indicate negative values). **b** Strength of the horizontal circulation in the Nordic Seas quantified as the absolute value of the minimum barotropic streamfunction in CESM-LE (63-76°N, 10°W-10°E). Solid line shows the ensemble mean and the shading indicates the interquartile spread. **c** Trends in horizontal circulation strength for CESM-LE, CESM-2C, CMIP6-SSP585 and CMIP6-SSP245. Trends are calculated for all 30-year periods between 2051-2100. Red circles show the mean value and the grey lines show the interquartile spread. **d** Ensemble trend correlation between all 30-year trends between 2006-2100 in Nordic Seas overturning strength (MOC$_\sigma$) and horizontal circulation in CESM-LE ($r = 0.76$, $n = 1980$).

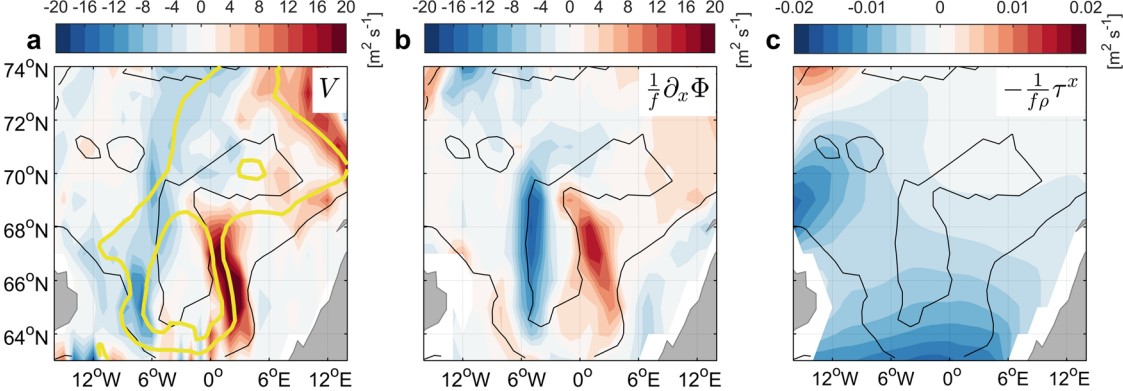

**Fig. 4 | Drivers of horizontal circulation changes in the Nordic Seas.**
**a** Differences in vertically integrated meridional transport in the eastern Nordic Seas between 2071–2080 and 2031–2040 (m$^2$ s$^{-1}$) in CESM-LE. Positive (negative) values on the eastern (western) side of the domain indicate strengthened cyclonic circulation. Yellow contours show the corresponding differences in the barotropic streamfunction (1 and 2 Sv isolines). Black lines are the 1000 m and 3000 m isobaths. **b**, **c** Vertically integrated meridional transport changes as computed from the potential energy term and wind stress, respectively (Equation (4)).

Variations in surface-forced water mass transformation can be driven by both air-sea fluxes and by the area covered by a particular surface density range[35]. Calculating MOC$_s$ using a seasonal climatology for buoyancy fluxes or surface density reveals that, while the decrease toward the 2040s is due to less buoyancy loss (Fig. 7c), changes in surface density explain the strengthening toward the end of the century (Fig. 7b). These density changes are driven by temperature. Consistent with an influence from large-scale hydrographic variability, as captured by the AMV index (Fig. 5a), future SST in the northern Nordic Seas (70–80°N) shows distinct multidecadal variations relative to

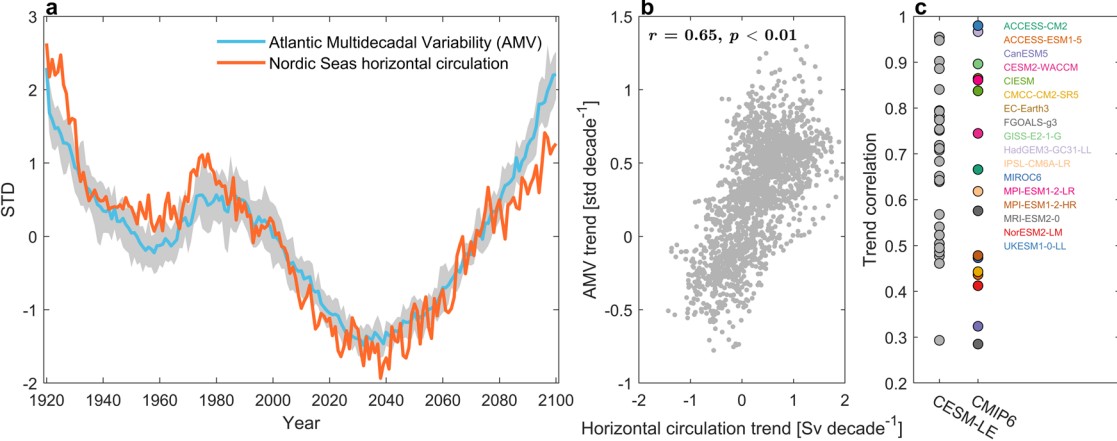

**Fig. 5 | Horizontal circulation and Atlantic Multidecacal Variability (AMV).**
**a** Standardized time series of horizontal circulation strength in the Nordic Seas and the AMV index in CESM-LE. For the AMV, solid line shows the ensemble mean and the shading indicates the interquartile spread. Only the ensemble mean is shown for the horizontal circulation. The AMV index is calculated from low-frequency component analysis[55,62] (Methods). **b** Ensemble trend correlation for all 30-year trends between 2006–2100 in Nordic Seas horizontal circulation and the AMV in CESM-LE ($r = 0.65$, $n = 1980$). **c** Correlations between 30-year AMV and horizontal circulation trends for individual ensemble members in CESM-LE (2006-2100) and for individual CMIP6 models (2015-2100; SSP585).

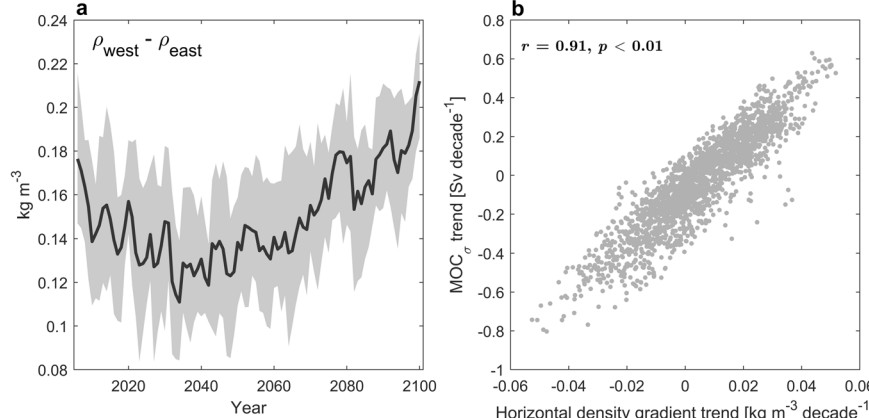

**Fig. 6 | Horizontal density gradient. a** West-east density gradient across 70°N between 200–500 m in CESM-LE. The gradient is calculated between 10–20°W and 7–17°E. Solid line shows the ensemble mean and the shading indicates the interquartile spread. **b** Ensemble trend correlation for all 30-year trends between 2006–2100 in Nordic Seas overturning strength ($MOC_\sigma$) and horizontal density gradient ($r = 0.91$, $n = 1980$).

to the linear trend (Supplementary Fig. 8). These results show that the increased water mass transformation necessary to sustain an enhanced $MOC_\sigma$ can, in part, be explained by surface forcing. Additional water mass transformation could also be happening due to increased mixing between the boundary current and the interior, e.g., as a result of increased horizontal circulation and baroclinicity[36,37]. This contribution is, however, not quantified here.

## Deep convection

Changes in deep convection could also influence the overturning circulation in the Nordic Seas[37,38]. In CESM-LE, deep convection – identified by mixed-layer depths (MLD) in March – takes place in the northern Nordic Seas (Supplementary Fig. 9). Maximum MLD in this region decreases until approximately 2060 and thereafter stabilizes. The reduced overturning toward the 2040s is thus consistent with reduced surface buoyancy fluxes and reduced convection. Considering 30 year trends for 2006–2100, the ensemble trend correlation between $MOC_\sigma$ and MLD is, however, low ($r = 0.22$) and not significant. Deep convection is thus clearly important in setting intermediate densities in the Nordic Seas, but there is no evidence that trends in deep convection are driving the deep density changes, and, hence, the

increased overturning toward the end of the century. We also note that no new regions of deep convection emerge in the Arctic Ocean during the 21st century in CESM-LE. However, if deep convection emerges in the Arctic Ocean in the future[17,18] this could become an important additional mechanism of overturning changes.

## Implications for overturning circulation in the subpolar North Atlantic

Changes in Nordic Seas overturning have the potential to influence the lower limb of the overturning circulation in the subpolar North Atlantic (SPNA) via the dense overflows across the Greenland-Scotland Ridge[2,3,9]. To represent the strength and properties of overflow waters across the Greenland-Scotland Ridge the CESM-LE uses an overflow parameterization[39]. The parameterized overflows through the Faroe-Shetland Channel (FSC) and Denmark Strait are driven by the density difference between the Nordic Seas source waters at sill depth and waters south of the ridge at sill depth. Following a parameterized entrainment of ambient downstream waters, the overflow (product) waters from the Nordic Seas are injected downstream of the Denmark Strait and FSC at the depth where neutral buoyancy is achieved (depths of 1500–3000 m). The mean parameterized overflow (source)

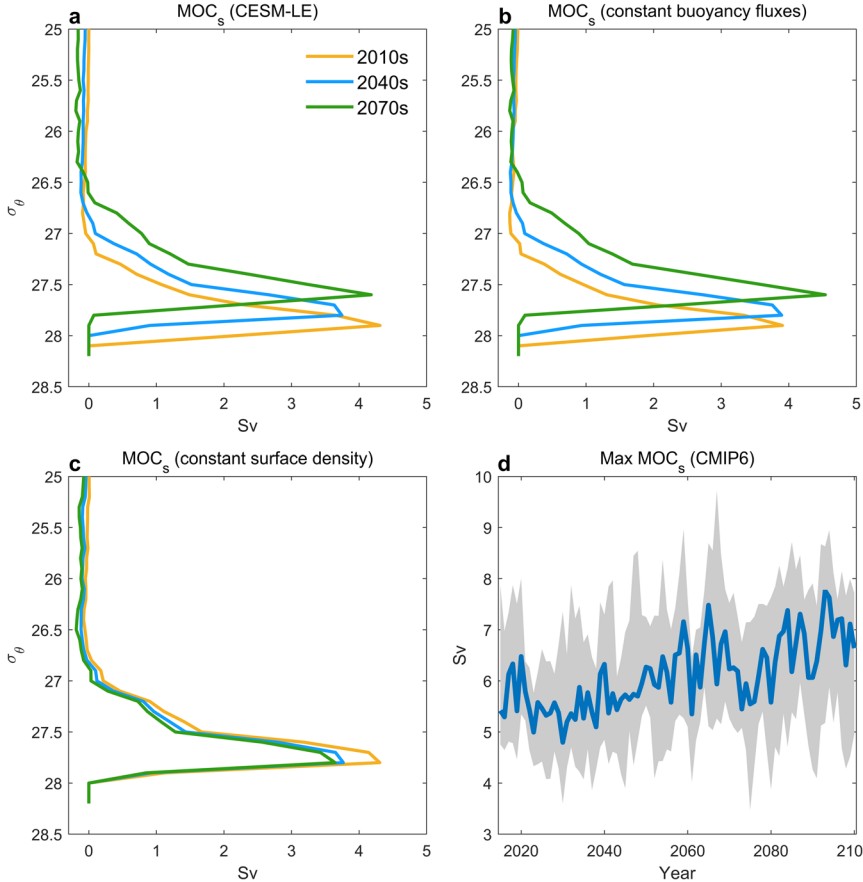

**Fig. 7 | Surface-forced water mass transformation. a** Decadal changes in surface-forced water mass transformation (MOC$_s$; Equation (3)) in CESM-LE calculated for the area 70-80°N, 35°W-20°E. The three lines show the ensemble mean for 2010–2019, 2040–2049, and 2070–2079. **b, c** As (**a**) but calculated using constant (seasonal climatology) buoyancy fluxes and constant surface density field, respectively. **d** Time series of maximum surface-forced water mass transformation in CMIP6 (16 models; Supplementary Table 1). The blue line shows the multi-model mean and the shading indicates the interquartile spread. Note that MOC$_s$ from CMIP6 models does not include freshwater flux into the ocean as the surface buoyancy flux is dominated by heat loss in this region.

transports from the FSC and Denmark Strait are 2.1 Sv and 2.7 Sv for 2011–2020, respectively, which are close to observational estimates[40]. Focusing on the trend reversal in the simulated Nordic Seas overturning strength in the 2040s, Fig. 8 indicates that the FSC overflow also shifts toward higher transports during this period (ensemble trend correlation of 0.56). In contrast, changes in Denmark Strait overflow show no significant relationship to Nordic Seas overturning (not shown). A stronger response in the FSC overflow than in the Denmark Strait overflow is consistent with the changes in horizontal circulation, showing a strengthening of the southward flow extending toward the FSC (Fig. 4a). The wind stress curl north of the FSC does not show trends consistent with Ekman pumping being responsible for the density changes in the Nordic Seas source waters. Any changes in the parameterized overflow in CESM-LE will also reflect density changes south of the Greenland-Scotland Ridge, whose origin are not investigated here. A detailed analysis of the circulation and entrainment of overflow waters in the SPNA is also not presented here. Our results nevertheless imply that overturning changes in the Nordic Seas could impact the overturning circulation in the SPNA via overflow changes in the FSC.

To quantify the overturning circulation in the eastern SPNA in CESM-LE we consider a section between Greenland and Scotland (Fig. 1) that closely resembles the eastern part of the array from the Overturning in the Subpolar North Atlantic Program[13] (OSNAP-East). The density-space overturning strength at OSNAP-East in CESM-LE is stronger than observed, but the overturning streamfunction is generally in good agreement with observations (Fig. 9a). The temporal

evolution of the simulated OSNAP-East overturning strength is characterized by fairly stable values until after the 2000s, after which there is a gradual decline in overturning strength in both depth- and density space (Fig. 9b). To assess whether the simulated overturning at OSNAP-East is impacted by overturning changes in the Nordic Seas we calculate the overturning streamfunction at OSNAP-East in density-space, but remap (for each year) the overturning streamfunction from density-space to depth-space using the zonal mean depth of each isopycnal. This is done in order to account for future changes in the density levels impacted by the Nordic Seas overflow waters. Trends in Nordic Seas overturning strength for each individual ensemble member are then correlated with trends in the (remapped) OSNAP-East overturning streamfunction for each depth level, considering all 30-year trends between 2006–2100. Figure 9c shows the relationship between trends in maximum Nordic Seas overturning and trends in the OSNAP-East overturning streamfunction at all depths. In support of Nordic Seas overturning changes manifesting at OSNAP-East we find significant correlations for depths corresponding to the Iceland-Scotland Overflow Water that is mainly fed by the Nordic Seas overflow waters via the FSC[39,41] (approximately 1500–2000 m). The relationship between Nordic Seas overturning trends and OSNAP-East overturning trends at 2000 m is further detailed in Fig. 9d and Fig. 9e, the latter examining whether the identified relationship depends on the trend length considered. The ensemble trend correlations decrease somewhat for shorter trend lengths (Fig. 9e), but remain significant for trends longer than 10 years. Consistent with these results, previous studies using the CESM and its predecessor CCSM4 have shown that changes in the

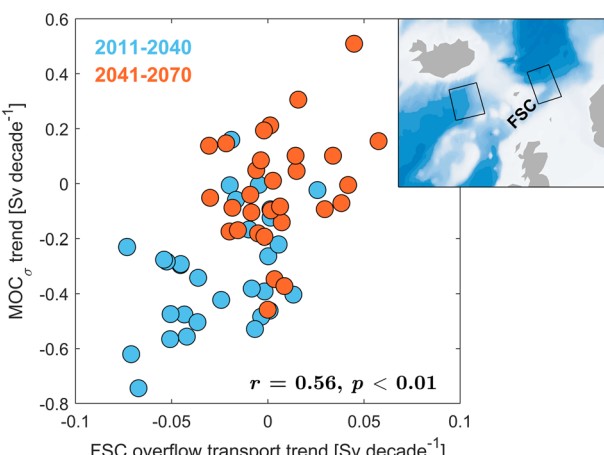

**Fig. 8 | Nordic Seas overflow.** Ensemble trend correlation between Nordic Seas overturning strength (MOC$_\sigma$) and overflow transport through the Faroe-Shetland Channel (FSC) in CESM-LE. Blue circles show 30-year trends for 2011–2040, while red circles show trends for 2041–2070. Considering both periods, the correlation ($r$) is 0.56 ($n = 60$). The regions used in the overflow parameterization[39] are shown in the inset map.

amount of (parameterized) overflow waters are associated with AMOC changes centered at 2000 m north of 60°N[42,43]. Our results, however, demonstrate for the first time that even if the overturning in the SPNA is expected to decline in the future, the magnitude of this decline is connected to changes in the Nordic Seas overturning circulation. It is important to note that the simulated changes in the Nordic Seas overturning circulation are reflected in the lower limb of the overturning circulation at OSNAP-East, but not in the maximum overturning strength which is found at shallower depths/lower densities[13].

We emphasize that, although the ensemble trend correlation between Nordic Seas and OSNAP-East overturning is significant, a variance explained of approximately 25% ($r_{max} \sim 0.5$; Fig. 9c) implies that other water masses than Nordic Seas overflow waters are also of major importance for subpolar overturning changes[6,44]. Trends in FSC overflow transport (Fig. 8) are also weaker than for the lower limb overturning circulation at OSNAP-East (Fig. 9d). Recent studies have suggested that surface buoyancy fluxes in the SPNA are a major driver of the lower limb overturning circulation at OSNAP-East[14,45]. However, ensemble trend correlations between simulated OSNAP-East overturning and net surface heat fluxes over the Labrador Sea and the Irminger Sea only show significant relationships in the upper 500 m (more heat loss associated with stronger overturning), and not at the depths associated with the Nordic Seas overturning circulation (1500–2000 m; not shown).

## Discussion

Using large ensemble simulations (CESM-LE) and CMIP6 models, we have shown that the overturning circulation in the Nordic Seas is projected to increase during the latter half of the 21st century as a result of enhanced horizontal circulation and a strengthened zonal density gradient. Simulated changes in the Nordic Seas overturning circulation show a clear link to large scale redistribution of sea surface temperatures that are captured by the AMV index. The modified surface density that results has implications for surface-forced water mass transformation in the Nordic Seas and the strength of the horizontal circulation. The future strengthening of the Nordic Seas overturning circulation is in contrast to overturning changes at lower latitudes which show a gradual future decline[19]. A more vigorous overturning circulation in the Nordic Seas furthermore acts to reduce the future overturning decline in the SPNA. The Nordic Seas are thus key to the state of the MOC[9].

Conclusions based on climate model simulations are only as reliable as the model's ability to correctly simulate the underlying processes. Climate models typically have a horizontal resolution in the ocean of 1° and therefore do not accurately resolve water mass transformation processes along the narrow boundary currents in the Nordic Seas[46,47]. Climate models also do not accurately simulate deep convection in the Greenland Sea[48]. These shortcomings could lead to model biases in horizontal density gradients and to an overestimated contribution of the horizontal circulation to the overturning circulation in density-space[12]. It has been shown that increased horizontal resolution in the ocean can have an impact on the mean state and long-term overturning trends in the North Atlantic[49–51], but such an analysis has not been performed for the Nordic Seas. The evaluation of the findings in this paper in high-resolution models is therefore warranted. However, we note that CMIP6 models with resolution higher than 1° (GFDL-CM4, GFDL-ESM4, MPI-ESM1-2-HR, and MRI-ESM2-0; Supplementary Table 1) do not stand out from the multi-model mean for the analyses performed here.

Despite the above-mentioned limitations, the CESM-LE has a realistic present-day Nordic Seas inflow of approximately 8 Sv[40] (Supplementary Fig. 5) which experiences an 8°C temperature decrease before returning as cold outflow, corresponding to a ∼ 300 TW heat loss within the Nordic Seas[9]. This translates into an overturning strength which is in close agreement with higher resolution (0.25°) ocean reanalysis products (Fig. 2), providing confidence in the ability of CESM-LE to simulate the overall transformation of light-to-dense waters in the Nordic Seas.

In conclusion, this study provides evidence that the overturning circulation in the Nordic Seas could be a stabilizing factor in a weakening North Atlantic overturning. These regionally dependent circulation changes in response to future climate change furthermore imply that current changes in the North Atlantic overturning[52] should not be extrapolated to the Nordic Seas and Arctic Ocean.

## Methods
### CESM-LE
To assess ocean circulation changes in the North Atlantic and Nordic Seas we use data from the Community Earth System Model Large Ensemble (CESM-LE[53]). The spatial resolution of the CESM ocean (POP2) and sea ice (CICE4) models is nominally 1° longitude by latitude, whereas the atmospheric model (CAM5) is 0.9° × 1.25°. The ocean model has 60 vertical levels. The CESM-LE has been widely used and extensively evaluated for the subpolar North Atlantic[28,54,55] and Nordic Seas[56,57].

The CESM-LE includes 40 ensemble members for the time period 1920 to 2100 using historical forcing for the period 1920–2005 and Representative Concentration Pathway (RCP[58]) 8.5 forcing from 2006 to 2100. Here we use the original 30 ensemble members[53] (#1-30). As the ensemble members all use the same Earth system model and the same external forcing, the ensemble mean represents the externally forced signal and the ensemble spread represents internal climate variability originating from the very small differences in the initial conditions of each member. Except where otherwise stated, all of our analysis is based on annual-mean fields.

To assess the sensitivity to future emission scenario, we also analyze a CESM ensemble that limits global warming to 2°C (CESM-2C[59]). This ensemble, which only differs from CESM-LE in the external forcing, consists of 11 members over the period 2006–2100.

### Calculation of overturning circulation
We calculate the strength of the meridional overturning circulation in the Nordic Seas at 70°N in both density space and depth space. Meridional velocities are integrated between the western and eastern boundaries and from the surface across depth and density layers for the depth- and density-space overturning streamfunctions,

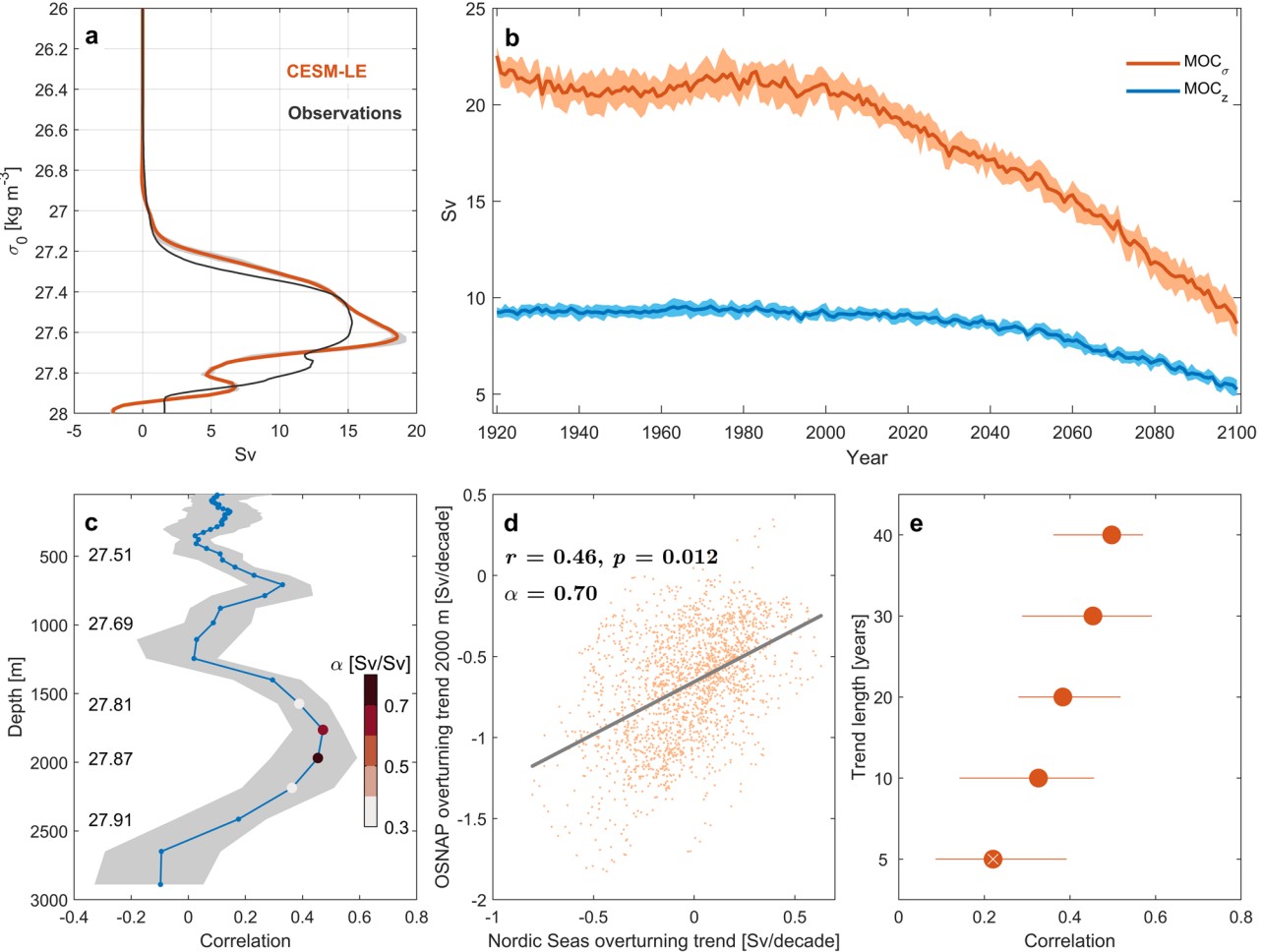

**Fig. 9 | Link between Nordic Seas overturning and the subpolar North Atlantic.**
**a** Simulated and observed[13] overturning streamfunction between 2014-2018 across the eastern OSNAP (Overturning in the Subpolar North Atlantic Program) section (OSNAP-East). **b** Time series of maximum overturning strength at OSNAP-East in depth- and density-space from CESM-LE. Solid line shows the mean value and the shading indicates the interquartile spread. **c** Correlation between trends in maximum Nordic Seas overturning strength and trends in the OSNAP-East overturning streamfunction (remapped from density-levels to depth-levels, see main text; ensemble mean density values for 2006–2100 are provided for some depths) for all 30-year periods between 2006–2100 ($n = 1980$). Solid line shows the mean value and the shading indicates the interquartile spread. Colored circles show the ensemble trend regression slope $\alpha$ for significant correlations at the 95% confidence level. **d** Relationship between Nordic Seas overturning trends and OSNAP-East overturning at 2000 m. The ensemble trend correlation ($r$) and regression slope ($\alpha$; OSNAP-E overturning trends regressed onto Nordic Seas overturning trends) are provided. **e** Ensemble trend correlations for different trend lengths for OSNAP-East overturning at 2000 m. Red circles show the mean value and the horizontal lines indicate the interquartile spread. White cross indicates non-significant correlation.

respectively. We apply no compensation to account for the net volume transport through the Nordic Seas (0.6 Sv averaged across ensemble members). Time series of overturning strength in depth ($MOC_z$) and density-space ($MOC_\sigma$) are then defined as the maximum stream functions across depth and density levels, respectively[12,13].

$$MOC_\sigma = \max\left[\int_{x_w}^{x_e}\int_{\sigma_{min}}^{\sigma} v(x,\sigma)\,d\sigma\,dx\right] \quad (1)$$

$$MOC_z = \max\left[\int_{x_w}^{x_e}\int_{0}^{z} v(x,z)\,dz\,dx\right] \quad (2)$$

Meridional velocities are calculated by regridding model variables $U$ and $V$ (Eulerian-mean, i.e., resolved), defined on the native curvilinear grid, onto a regular 1° longitude-latitude grid. Densities are regridded onto the same grid. We have made sure that this regridding does not influence our results. We have also assessed whether the changes seen in the overturning strength are sensitive to the chosen

latitude by calculating the overturning circulation for a section on the native model grid that cuts across the southern Nordic Seas (Supplementary Fig. 3). Considering 30-year trends in $MOC_\sigma$ between 2006–2100, the correlation between these two sections is 0.93 (not shown). This demonstrates that, in CESM-LE, 70°N is a representative latitude in terms of overturning changes occurring in the Nordic Seas.

The overturning in the SPNA is calculated on the native model grid along a section between the southern tip of Greenland and northern Scotland (Fig. 1), closely resembling the OSNAP-East monitoring array[13]. The overturning profile from CESM-LE is in good agreement with observations from OSNAP (Fig. 9a).

**Surface-forced overturning circulation**
The surface-forced overturning streamfunction ($MOC_s$; in units of Sv) across an isopycnal, $\sigma$, was calculated as[14,34]:

$$MOC_s(\sigma^*) = \frac{1}{\Delta\sigma}\int\int_{dA}\left[-\frac{\alpha Q_H}{C_p} + \beta\frac{S}{1-S}Q_{FW}\right]\Pi(\sigma)\,dA, \quad (3)$$

where

$$\Pi(\sigma) = \begin{cases} 1 & \text{if } \sigma - \Delta\sigma/2 < \sigma < \sigma + \Delta\sigma/2 \\ 0 & \text{elsewhere} \end{cases}$$

$\alpha$ is the thermal expansion coefficient, $\beta$ is the haline contraction coefficient, $C_p$ is the specific heat capacity of seawater, $Q_H$ net surface heat flux into the ocean, $S$ is surface salinity, and $Q_{FW}$ is the net freshwater flux into the ocean. $\text{MOC}_s$ is calculated for each month and each isopycnal $\sigma$ (spaced by $\Delta\sigma = 0.1\,\text{kg m}^{-3}$) and then averaged into annual fields. If $\sigma$ do not outcrop within the defined region in a given month, $\text{MOC}_s$ is set to zero. As surface buoyancy loss is dominated by ocean heat loss to the atmosphere, $\text{MOC}_s$ from CMIP6 models is calculated using only $Q_H$.

### Ensemble trend correlations
To assess the mechanisms of and impacts from changes in Nordic Seas overturning we analyze the spread of trends across the ensemble, that is, we calculate correlations with and regressions onto the 30 linear trend values of Nordic Seas overturning obtained from the CESM-LE. This is referred to as ensemble trend correlation (regression)[60]. As the externally forced trend is the same for each ensemble member these ensemble trend regressions and ensemble trend correlations describe relationships between internal variability in the trends of various fields and the internal variability in Nordic Seas overturning trends. Statistical significance levels are computed based on a random phase test[61].

### Low-frequency component analysis
To identify multidecadal variability in North Atlantic SST we use low-frequency component analysis[62] (LFCA). LFCA isolates the low-frequency variability in a dataset by finding low-frequency patterns that maximize the ratio of low-frequency to total variance in their corresponding time series, which are called low-frequency components (LFCs). For a detailed description see ref. 62. LFCA has previously been applied to identify decadal and multidecadal variability in the North Atlantic[55,62]. Following previous studies, we define low-frequency variance based on a linear Lanczos filter with a 10-yr low-pass cutoff. We use 10 EOFs.

The LFCA is based on annual mean SST for the area 0-60°N, 0-80°W and for the period 1920–2100 for CESM-LE and 2015–2100 for CMIP6 models. For CESM-LE, the analysis is performed for each ensemble member. For each ensemble member in CESM-LE or CMIP6 model, the first low-frequency component captures the global warming trend, whereas the second mode shows pronounced multidecadal variability. This second mode is referred to as Atlantic Multidecadal Variability (AMV). The time series of the first two LFCs in CESM-LE and their associated spatial patterns are shown in Supplementary Fig. 10.

### Gyre dynamics
To assess the dominant driver of gyre changes in the Nordic Seas we consider the depth-integrated volume transport equation in the meridional direction[24,63]:

$$fV = -H\partial_x P_b + \partial_x \Phi - \tau_x/\rho_0, \tag{4}$$

where $V$ is the vertically integrated meridional transport, $\tau_x$ is the zonal wind stress, $\rho_0$ is a representative density for sea water, $f$ the Coriolis parameter, $H$ the local water depth, and

$$\Phi = g \int_{-H}^{0} \frac{\rho_0 - \rho}{\rho_0} z \, dz \tag{5}$$

represents the potential energy of the water column and reflects the depth-averaged density stratification. The potential energy term is sensitive to density changes in deep water because of the weighting of $z$ in the calculation of $\Phi$. $P_b$ is the bottom pressure. However, this term is not considered here since an amplification of the flow can only result from the ageostrophic contributions from the potential energy and the wind stress terms[63]. Derivatives are calculated using centered differences on data regridded onto a regular 1° longitude-latitude grid. We note that different methods exist to calculate $\Phi$ (e.g., ref. 64), but that these produce qualitatively similar results (Supplementary Fig. 11).

### CMIP6
Results from CESM-LE are compared with those from CMIP6 models[65]. We use historical (1920-2014) and future (2015-2100) simulations using the Shared Socioeconomic Pathways[66] (SSP) scenarios SSP245 and SSP585. Overturning circulation in density space (model variables *msftmrho* and *msftyrho*) is not provided by many CMIP6 models. For SSP585, these variables were available from six models from three modelling centers at the time of download (Supplementary Table 1). For the barotropic streamfunction (BSF; *msftbarot*), we analyze 17 models that had output available for both scenarios (Supplementary Table 1). As the different models calculate the BSF in different ways, we calibrate the BSF by removing the mean BSF from the North Atlantic (70°W-50°E, 0-60°N). This ensures that positive (negative) values correspond to anticyclonic (cyclonic) gyres (Supplementary Fig. 12). Note that this calibration does not influence the magnitude of the trends that are the focus of this study. We also analyze SST (*tos*), SSS (*sos*), and net surface heat fluxes (*hfds*) from SSP585.

### Ocean reanalyses
To evaluate the strength of the Nordic Seas overturning circulation in CESM-LE and CMIP6 models, we use four commonly used ocean reanalyses; ORAS5[67] C-GLORSv7[68], GLORYS2V4[69], and GloSea5[70]. The horizontal resolution is 1/4° × 1/4° (approximately 12 km in the Nordic Seas) and there are 75 vertical levels (level spacing increasing from 1 m at the surface to 200 m in the deep ocean). The reanalyses covers the period 1993 to 2020. Although all of these ocean reanalyses are based on the NEMO ocean model with the same horizontal and vertical resolution, they show different mean states and variability in e.g., overturning strength in the North Atlantic[71]. The overturning streamfunction for each reanalysis is shown in Supplementary Fig. 2.

## Data availability
All the datasets used in this study are publicly available: CESM-LE data from   http://www.cesm.ucar.edu/projects/community-projects/LENS; CESM-2C   from   https://www.earthsystemgrid.org/dataset/ucar.cgd. ccsm4.lowwarming.html; CMIP6 data from the Earth System Grid Federation (ESGF) at https://esgf-node.llnl.gov/search/cmip6/; OSNAP observations   from   http://www.o-snap.org/observations/data/.   Data from ORAS5, C-GLORSv7, GLORYS2V4, and GloSea5 are available from https://resources.marine.copernicus.eu/product-download/GLOBAL_ REANALYSIS_PHY_001_031.

## Code availability
The Matlab code used to analyze and plot the data displayed in this paper is available from the corresponding author upon request. The code for the LFCA can be downloaded from https://github.com/ rcjwills/lfca.

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

## Acknowledgements

This study received support from the Bjerknes Centre for Climate Research (project DYNASOR; M.Å., H.A., K.V.), the Trond Mohn Foundation (project number BFS2018TMT01; MÅ), the UK Natural Environment Research Council under the SNAPDRAGON project (NE/T013494/1; H.L.J.), the UK Natural Environment Research Council grant U.K.-Overturning in the Subpolar North Atlantic Program Decade (NE/T00858X/1; L.C.), and the Swedish National Space Agency through the FiNNESS project (Dnr 133/17; L.C.) and the ECO2NORSE project (Dnr 2022-00172; L.C.). We thank the CESM Large Ensemble Community Project for making their data publicly available. We also acknowledge the World Climate Research Programme Working Group on Coupled Modelling, which is responsible for CMIP, and we thank the climate modeling groups for producing and making available their model output. OSNAP data were collected and made freely available by the OSNAP (Overturning in the Subpolar North Atlantic Program) project and all the national programs that contribute to it.

## Author contributions

M.Å. initiated the study, did the analysis, produced the figures and wrote the initial manuscript. H.A. provided and analyzed CMIP6 data. L.C. made Fig. 1b. M.Å., H.A., L.C., H.L.J., and K.V. contributed ideas, discussed the results, reviewed and edited the paper.

## Funding

## Competing interests

The authors declare no competing interests.
