## [Peer Review File · Nature Communications]

Future strengthening of the Nordic Seas overturning circulationRe: “Future strengthening of the Nordic Seas overturning circulation” by Årthun et al.

This presented analysis discusses future changes in the Nordic Seas overturning and explores its cause and effect. Based on climate model output, it shows opposing overturning changes between the Nordic Seas and the North Atlantic basins. The overturning variability in the Nordic Sea is attributed to the horizontal circulation strength, and such Nordic Sea changes are suggested to impose negative feedback on the North Atlantic overturning variability.

The topic of this manuscript is important for understanding the AMOC variability and its latitudinal coherence under climate change and will be of wide interest to the readership of NC. However, there are some important aspects of the inference that need attention. That leads to several concerns I have about this manuscript which are detailed below. Mainly, I found the discussion about the cause and effect of the Nordic Sea overturning not convincing, and thus cannot recommend the manuscript to be accepted for publication. I am a bit worried that even a revision in its present form may not be sufficient because this short-paper itself could be of insufficient length for the material the authors want to convey. But I will leave it for the editor to decide.

Major concerns:

1. The authors attribute the Nordic Seas overturning changes to the horizontal circulation, not the strength of deep-ocean convection (as indicated by max MLD in the region). But how robust the proposed linkage over the time scale of interests remains unclear. Comparing Figure 2 and Figure 3b, such a linkage between the horizontal circulation and overturning doesn't seem to exist before 2000. Even after 2000, the linkage may not exist in a consistent way. There is a more (less) rigorous overturning (horizontal circulation) change in the first-half of the 21st century compared to the second half.
2. The authors go on to suggest that the horizontal circulation changes are caused by local density anomalies in the Nordic Seas. Here the exclusion of an impact from deep convection may be hasty. How to exclude the role of deep convection in creating those density anomalies? Density anomalies could affect the overturning circulation directly, e.g., by modifying the vertical velocity shears. Has this mechanism been examined? The dramatic Nordic Seas overturning changes in the first-half of the 21st century could be related to the convective changes (comparing Figure 2 and Supplementary Figure 3). This relationship may or may not be as strong as that between the horizontal circulation and overturning.
3. The Nordic Seas overturning is known to form and export overflow waters, which are the densest waters within the lower limb of the subpolar overturning. Thus, it is not surprising that “overturning changes in the Nordic Seas could impact the overturning circulation in the SPNA” (lines 129-130). Unfortunately, the analysis doesn't show clear evidence for such coherence in

the overturning changes. In contrast, there are actually opposing overturning changes between the Nordic Seas and the North Atlantic region, e.g., after 2050 (e.g., lines 6-7, Figures 2 and 6).

4. A related question is really about the role of the overflow change at the Faroe-Shetland Channel (FSC), which is suggested to be the sole pathway connecting the Nordic Seas and subpolar overturning changes. However, the FSC transport trend is only approx. 1/10 of that of the NS overturning trend (Figure 5) and is even a smaller fraction of the subpolar overturning trend (Figure 6). How could such a small FSC change explain greater overturning changes? A more thorough evaluation is thus required on how this overflow transport contributes to the subpolar overturning variability. Otherwise, the results could imply a more significant impact from the Nordic Sea overturning on the exchange with the Arctic than that with the North Atlantic.

5. The Nordic Seas overturning is defined from the streamfunction at 70N at an arbitrary depth such as 2000m. How does the overturning strength change when calculated in a different way (e.g., maximum streamfunction at the GSR)? How representative this latitude is in terms of the overturning changes occurring in the Nordic Seas? There is a large basin between 70N and the Greenland-Scotland-Ridge (GSR), would that contribute to the overflow change discovered at the FSC? If an arbitrary depth is deeper than where the maximum is, it reflects overturning changes in both upper limb and partial lower limb, then what does it mean physically? Since a new definition of the overturning circulation strength is used, it requires more descriptions and validations.

6. Why is MOCz also presented if “overturning ... is best quantified by the meridional overturning streamfunction in density space” (lines 22-24)?

7. Lines 80-82: But as we know, some of the inflow of the North Atlantic join the Nordic Seas horizontal circulation and some participate in the overturning circulation. Thus, the change in the inflow may not be directly linked to the strength of the horizontal circulation.

8. Line 123: The mean overflow transport is close to observation but the subpolar overturning is larger than that observed. How does it affect the analysis that is focused on the relation between OW and overturning?

9. Lines 162-163: Water mass transformation in the Labrador and Irminger Seas cannot be expected to impact the transport variability in the overflow water layers.

10. Figure 3a: Please label the x- and y-axis.

REVIEWER COMMENTS

Reviewer #2 (Remarks to the Author):

Review of: "Future strengthening of the Nordic Seas overturning circulation", by Årthun et al.

In this paper, the authors analyze a strengthening of the overturning circulation in the Nordic Seas in the CESM Large Ensemble (CESM-LE) and other models from the CMIP6 archive, in the latter half of the 21st century. The authors link the circulation changes to changes in the density structure of the Norwegian Sea, which they attribute to changes in the wind stress curl over the region.

I believe there is a lot of merit in this paper, and I would love to see it published in some form. The future evolution of the Atlantic meridional overturning circulation (AMOC) is an important topic, and the role that the Nordic Seas (and Arctic in general) play in the AMOC definitely warrants more study. The finding that the overturning in the Nordic Seas may increase in the second half of the 21st century is novel and important, and may change our understanding of both the evolution of the AMOC at lower latitudes, as well as of the Arctic.

Still, there are some issues that, if addressed, would strengthen the paper. I would recommend the paper to be sent back to the authors for major revisions.

My primary concern is the following: even though the 'overturning circulation' in the Nordic Seas can be interpreted as horizontal circulation across sloping isopycnals, this in itself is only a kinematic statement. It does not recognize that flow through a stationary distribution of sloping isopycnals necessarily reflects a balance between advective and mixing processes. A strengthening of the horizontal circulation alone will not lead to enhanced overturning; it is only part of the story. In my mind, the question how the water mass transformations change is equally important and is not being addressed in this paper. I think an analysis of the water mass transformation processes that sustain the enhanced overturning in the second half of the 21st century would be essential to complete this study.

A second concern is that the analysis of the drivers of the changes in overturning was not very convincing to me. In particular the explanation that isopycnals in the western Norwegian Sea (below 1000 m nonetheless) are depressed in response to a localized change in wind stress curl and associated Ekman pumping did not convince me. It seems to me that some larger-scale adjustment of the circulation and stratification must be involved. Maybe an analysis of additional CMIP6 models, or an analysis of intra-model spread in CESM-LE, could yield some more convincing evidence for the mechanism proposed, or maybe provide additional insights. Maybe an analysis of water mass transformations might help here as well.

A third concern is that it should probably be emphasized more that this is a model analysis study. This should at least be mentioned in the abstract, if not in the title also. Also, the analysis is based mostly on a single model configuration. It is encouraging that the few available CMIP6 models support the strengthening of the overturning as found in CESM-LE (Fig. 2). But 6 models are not many, and no ensemble spread is shown in Fig. 2. What's more, Fig. 3c shows that CESM-LE is significantly higher in another metric (gyre strength) than the CMIP6 ensemble mean. One option would be to expand the CMIP6 ensemble by not just focusing on the models that provided overturning in density coordinates as standard output, but to reconstruct this overturning from (regridded) velocity and density fields (as is being done for the CESM-LE). In addition, I'm wondering if any high-resolution (HighResMIP?) simulations could be included to expand the ensemble. It would also help to also show the historical period for the CMIP6 ensemble in Fig. 2, to provide some better insights in the degree of agreement of the CMIP6 models and CESM-LE over longer time scales. Are there any observations or

reanalyses products (e.g., RARE, or ASTE) that could provide some validation for the CESM-LE historical behavior? As it stands, the basis for the conclusions is rather thin.

Below is a list of minor and major concerns.

I. 4, 19: Nordic Seas is -> Nordic Seas are (?)

I. 27: In my mind the statement that density-space overturning is dominated by horizontal circulation across sloping isopycnals is only a kinematic statement. It still requires mixing processes to balance advection in order to maintain sloping isopycnals. It may be my limited understanding, but to me it seems that processes like open deep convection are still critical for water mass transformations; I therefore don't see how it challenges the importance of open deep convection.

L. 37: I don't think it is emphasized enough that the conclusions are based on an analysis of numerical models. It should at least be mentioned in the abstract, if possible, in the title, and definitely in the first paragraph of the section starting at l. 37.

I. 44: I understand that there are only a limited number of CMIP6 models that have saved overturning in density space. Given that the main analysis is done on CESM-LE, for which the overturning is reconstructed from density and velocity fields (regridded, nonetheless), there is not really an excuse to not calculate this quantity for more models.

ll.69-70: I think here the importance of water mass transformations is neglected. How are the sloping isopycnals maintained in the face of a strengthening gyre circulation?

I. 72: How realistic is this distribution of MLD in CESM-LE, compared to observations?

I. 103: I'm not convinced about this argument. It's hard to imagine that the deepening of the isopycnals below 1000 m can be attributed to locally enhanced Ekman pumping. I like the effort to quantify the impact of enhanced Nordic Seas overturning on overturning in the subpolar North Atlantic, I think this is well done.

I. 173: It's not clear why the analysis is limited to only 30 ensemble members, when 40 are available?

I. 188: I would be nervous calculating transport-based quantities like overturning strength based on regridded fields. The authors mention there is no compensation applied to account for Bering Strait throughflow, but is it clear that any errors introduced by regridding are negligible?

Fig. 1: Please include lat/lon axis labels.

Fig. 2: I think it would be useful to show CMIP6 simulations for the historical period as well.

Fig. 3: Please include lat/lon axis labels. Maybe indicate the location of the minimum barotropic streamfunction, from which panel b is extracted. I would recommend extending the domain in panel a northward to include the entire Nordic Seas. I think it would be useful to reiterate that most panels show CESM-LE results.

Fig. 4: If the MOC at 70N is the subject of this paper, it is not clear to me why the authors show transects at 67N instead. The main transport features that the analysis focuses on extend only to 69N, so it's hard to see how this explains the MOC behavior further north.

Fig. 5: Again, please reiterate that these results are from CESM-LE. Why are trends for the period 2041-2070 shown here? Why not for 2015-2100, for instance, as in Fig. 3d?

Table 1: Modeling centers really like to be acknowledged for their efforts. One way authors can do this is by citing relevant model description papers, or by explicitly mentioning modeling centers in the acknowledgements. But most importantly, all CMIP6 model output now have citable doi's, and one can pull up references on ESGF. I know it is incredibly tedious to assemble this information, especially when the number of data sets used approaches many tens, or even hundreds. But I believe it is a responsibility we have; and publishers should support this too.

-Wilbert Weijer

Reviewer #3 (Remarks to the Author):

In this paper, the authors use the Community Earth System Model Large Ensemble

(CESM LE) to predict that the meridional overturning circulation in the Nordic Seas will increase in the second half of the 21st century. This contrasts with predictions that the MOC in the subpolar and subtropical North Atlantic will decrease. By analyzing the model output, the authors further attribute the increase to a strengthening of the cyclonic circulation in the Nordic Seas driven by changes in wind stress curl. Finally, they argue that these high-latitude changes may partially offset the predicted decline in overturning in the subpolar North Atlantic. The results are interesting and at a high level, would be of interest to physical oceanographers and ocean climate scientists. Before I add my comments, I want to disclose that I'm not a subject-matter expert in numerical modeling, so I'm not in a position to evaluate the choice of model nor how this study fits within other literature that uses numerical model simulations to make inferences about past or future changes in the Atlantic meridional overturning circulation (AMOC). I'm an observational physical oceanographer involved in AMOC research who (I think) can evaluate the general methods and interpretations made in this paper and how they might be understood by non-specialists.

This perspective informs my first general comment: the authors must be more up front about (a) the fact that this study is based entirely on numerical model simulations, and (b) the pros and cons of these coarse-resolution coupled climate models for applications at high latitudes. With respect to (a), the issue starts right in the abstract with "We show that...the overturning circulation in the Nordic Seas increases..." In my opinion, this should read something like, " We show that simulated overturning circulation..." or something similar. Similarly, the first sentence in the section "Future Increase in Nordic Seas Overturning Circulation" starts out "Time series of the strength..." should be written something like "Model time series..." I realize that the focus of this study on future changes implies that some kind of model is being used, but The authors also show model output from the past, so I feel strongly that any results should be qualified and identified explicitly as being from a model, at least in the first part of the paper. As for (b), I also believe the authors should be more descriptive about the model used and its pros and cons. To their credit, they have one sentence just before section "Implications of Overturning Circulation in the Subpolar North Atlantic" noting that because of its coarse resolution, the model may over-emphasize the role of increased circulation compared to smaller-scale boundary processes, but this is not really adequate. In fact, I don't think the authors say anywhere what the spatial resolution of the model is, and how this scale compares to the natural scale of variability (the deformation radius; which is very small at these high latitudes). The authors must include this kind of basic model description if they want to publish their results in a general-audience scientific journal like Nature Communications.

Finally, the figures are very dense with information and in my opinion, some are challenging to unpack, especially for someone who is not an expert in numerical modeling. The authors have been creative in layering variables in order to minimize the number of figures, but this means one really has to spend a lot of time studying the figures and their captions to understand the message. It seems to me that the authors' target audience is the modeling community in which they are embedded. Maybe this is ok for Nature Communications-I leave this up to the editor to decide.

Otherwise, the manuscript is well-written and the figures are of generally good quality (aside from comment above about complexity, and some specific figure comments below).

Specific comments:

Line 6 - This phrase demonstrates one of my primary concerns about this paper, which is that the authors do not indicate front and center that this claim ("we show that [the MOC of the Nordic Seas] increases...") is based on output from a coarse-resolution coupled climate model (see general comment above).

Line 16 - What do the authors mean by "here" in this sentence? The next sentence implies they mean the Nordic Seas and Arctic, but in fact the transformation occurs also in the North Atlantic.

Line 19 - "is" should be "are"

Line 27 - Should consider referencing Petit and Koman papers here also.

Line 32 - Need to get prediction in here somewhere.

Line 37 - The authors must state the source of the time series (see general comment above). It is one thing to lean on the "Methods" section to keep the word count in the main body of the paper within limits, but I believe this is a section where the phrase "time series" should be qualified as "Model time series".

Line 54 - The authors attribute the predicted strengthening of the MOC in density space to increase in strength of the circulation. Couldn't it also be explained by an increase in the horizontal density gradient across which the flow is oriented?

Line 57 - Since the direction of circulation in the subpolar North Atlantic and Nordic Seas is cyclonic, and the two regions are connected, couldn't the pattern observed be described as a shift of the center of the joint cyclonic circulation northward?

Line 63 - The authors say there is a large spread in future trends of Nordic Seas circulation, which could be contradictory to the first part of the sentence where they say all the models show an increase. This sentence needs to be made more clear.

Line 70 - This addresses somewhat the question of uncertainty in whether it's increased circulation or increased horizontal density gradients.

Line 84 - Are these two mechanisms independent? Wind stress curl has a direct influence on density structure through Ekman pumping/suction. The authors make this very point later. But the way this introductory sentence is phrased, it sounds like the two mechanisms are independent. This whole section needs clarification, to better differentiate between effects of wind stress, wind stress curl, changes in density structure (due to wind stress curl), etc.

Line 89 - I don't think "density-driven transports" is the right phrase here. It is transport changes driven by changes in the structure of the density field. Density-driven transport sounds to me like a gravity current or deep convection.

Line 93 - Is this Ekman transport?

Line 100 - At first blush, it doesn't make sense to me that a deepening of isopycnals in the west relative to the east would lead to a stronger cyclonic circulation--quite the opposite I would think since this sounds like a relaxation of the domed isopycnals that balance the cyclonic geostrophic flow. So a deepening of isopycnals in the west and a shoaling in the east seems like would result in a weaker northward meridional transport.

Line 128 - Could the increase in FSC transport also be caused by the shoaling isopycnals in the eastern Nordic Seas, which would mean higher densities at the upstream end of the channel at the sill depth?

Line 193 - How small is this transport?

Figure 1 - Please make the OSNAP line more visible in lower panel. Label panels a and b to match caption.

Figure 1 - Please add latitude and longitude labels so as to better place locations of time series described in future figures.

Figure 2 - is there observational evidence of decrease in MOC between 2005-2022 in Nordic Seas? This would lend confidence to the model results.

Bergen, 18.12.2022

ANSWER TO THE REVIEWER COMMENTS

We wish to thank the three anonymous reviewers for constructive comments and suggestions. All issues have to the best of our knowledge been addressed in the revised manuscript. Our response to the reviews is marked using blue font.

Best regards, on behalf of the authors,

Marius Årthun

Reviewer #1 (Remarks to the Author):

Re: "Future strengthening of the Nordic Seas overturning circulation" by Årthun et al.

This presented analysis discusses future changes in the Nordic Seas overturning and explores its cause and effect. Based on climate model output, it shows opposing overturning changes between the Nordic Seas and the North Atlantic basins. The overturning variability in the Nordic Sea is attributed to the horizontal circulation strength, and such Nordic Sea changes are suggested to impose negative feedback on the North Atlantic overturning variability. The topic of this manuscript is important for understanding the AMOC variability and its latitudinal coherence under climate change and will be of wide interest to the readership of NC. However, there are some important aspects of the inference that need attention. That leads to several concerns I have about this manuscript which are detailed below. Mainly, I found the discussion about the cause and effect of the Nordic Sea overturning not convincing, and thus cannot recommend the manuscript to be accepted for publication. I am a bit worried that even a revision in its present form may not be sufficient because this short-paper itself could be of insufficient length for the material the authors want to convey. But I will leave it for the editor to decide.

We thank the Reviewer for constructive comments and suggestions. Please find below our detailed response to each of the points raised.

In response to the concerns raised by the reviewers, we have substantially revised the manuscript and extended the analysis of the future overturning changes in the Nordic Seas. This includes 3 new figures in the main manuscript and 5 new figures in the supplementary material. The new analysis is especially concerned with the cause of Nordic Seas overturning changes. While further analysis would undoubtedly be revealing, we believe the presented analysis is sufficient to provide the necessary evidence in support of our conclusions.

It unfortunately seems that we did not write sufficiently clearly about the link between Nordic Seas overturning and OSNAP-East (comments 3-4), and how we calculate the overturning strength (comment 5), which has led to misunderstandings. We have worked hard to improve the manuscript so that these points are clearer (see our detailed replies below).

Major concerns:

1. The authors attribute the Nordic Seas overturning changes to the horizontal circulation, not the strength of deep-ocean convection (as indicated by max MLD in the region). But how robust the proposed linkage over the time scale of interests remains unclear. Comparing Figure 2 and Figure 3b, such a linkage between the horizontal circulation and overturning doesn't seem to exist before 2000. Even after 2000, the linkage may not exist in a consistent way. There is a more (less) rigorous overturning (horizontal circulation) change in the first half of the 21st century compared to the second half.

To assess the robustness of the linkage between the horizontal overturning circulation and Nordic Seas overturning, we have calculated the ensemble trend correlation (30-year trends as in Figure 3d) between MOC and gyre for different periods in both the historical and future simulations (see figures below). We find no systematic differences in correlation between different time periods, showing that the relationship is robust. We have added to the text (l.82-83): "The correlations are robust for different periods, e.g., considering the first and second part of the 21st century, and for the historical simulation (1920-2005)."

Note that we realized upon investigating the linkage between the horizontal overturning circulation and Nordic Seas overturning in more detail that we were calculating the circulation strength (minimum barotropic streamfunction) for a smaller region than that stated in the caption of Fig.3 (using a northern boundary of 70N instead of 76N). This has now been corrected. This does not change the correlations presented in Figure 3 or any of our findings in this paper but changes the historical time series somewhat, making it more similar to the overturning time series.

Figure: (left) 30-year trends in overturning circulation and horizontal circulation for the first and second part of the 21st century. Regression slopes are shown for the first (black line) and second half (blue line) of the time series. (right) Ensemble trend correlation for the historical simulation.

2. The authors go on to suggest that the horizontal circulation changes are caused by local density anomalies in the Nordic Seas. Here the exclusion of an impact from deep convection may be hasty. How to exclude the role of deep convection in creating those density anomalies? Density anomalies could affect the overturning circulation directly, e.g., by modifying the vertical velocity shears. Has this mechanism been examined? The dramatic Nordic Seas overturning changes in the first half of the 21st century could be related to the convective changes (comparing Figure 2 and Supplementary Figure 3). This relationship may or may not be as strong as that between the horizontal circulation and overturning.

Deep convection is clearly important in setting intermediate densities in the Nordic Seas, but there is no evidence that *trends* in deep convection are driving overturning changes. To further quantify the relationship between MOC and MLD we have calculated the ensemble trend correlation for all 30-year trends in CESM-LE (same as that done for horizontal circulation vs overturning; Figure 3d). The resulting correlation is only 0.22 for the period 2006-2100. This has been added to l.172-174.

To further strengthen our analysis of changes in the overturning circulation, we have calculated the surface-forced water mass transformation, which relates the rate of density transformation in a given density class to the surface buoyancy fluxes into that density class over its outcrop area (e.g., Marsh 2000). This shows that reduced buoyancy loss (consistent with reduced convection) was a main driver of reduced overturning between 2010s and 2040s. This has been added (Fig. 5; l.139-166). We note, however, that the water mass transformation analysis suggests that the increased overturning toward the end of the century is a result of surface density changes, and not surface buoyancy loss (l.156-159).

3. The Nordic Seas overturning is known to form and export overflow waters, which are the densest waters within the lower limb of the subpolar overturning. Thus, it is not surprising that “overturning changes in the Nordic Seas could impact the overturning circulation in the SPNA” (lines 129-130).

Unfortunately, the analysis doesn't show clear evidence for such coherence in the overturning changes. In contrast, there are actually opposing overturning changes between the Nordic Seas and the North Atlantic region, e.g., after 2050 (e.g., lines 6-7, Figures 2 and 6).

It is true that the Nordic Seas and SPNA show opposing long-term changes in overturning circulation (Figures 2 and 9). However, as noted in the manuscript (l.226-227): "even if the overturning in the SPNA is expected to decline in the future, the magnitude of this decline is connected to changes in the Nordic Seas overturning circulation". This is clearly shown in Figure 9c, which compares 30-year trends in overturning; a stronger Nordic Seas overturning corresponds to a weaker MOC decline in the SPNA (i.e., positive correlation). This relationship is further quantified in Figure 9d,e. Hence, we do not agree with the reviewer that we do not provide evidence of a link between overturning in the Nordic Seas and in the SPNA. We also note that this analysis was complimented by Reviewer 2, and we have therefore not made any changes here.

4. A related question is really about the role of the overflow change at the Faroe-Shetland Chanel (FSC), which is suggested to be the sole pathway connecting the Nordic Seas and subpolar overturning changes. However, the FSC transport trend is only approx. 1/10 of that of the NS overturning trend (Figure 5) and is even a smaller fraction of the subpolar overturning trend (Figure 6). How could such a small FSC change explain greater overturning changes? A more thorough evaluation is thus required on how this overflow transport contributes to the subpolar overturning variability. Otherwise, the results could imply a more significant impact from the Nordic Sea overturning on the exchange with the Arctic than that with the North Atlantic.

First, we would like to emphasize that the trends in Nordic overturning and SPNA overturning are of the same magnitude (Figure 9c). Second, it is widely established that the transport of dense overflow waters from the Nordic Seas impacts the overturning circulation in the North Atlantic (as the reviewer states in the comment above; "the Nordic Seas overturning is known to form and export overflow waters, which are the densest waters within the lower limb of the subpolar overturning"). For the CESM (CCSM4), dedicated simulations and analysis have demonstrated that overflow variability impacts the AMOC at lower latitudes (Yeager & Danabasoglu 2012; Danabasoglu et al. 2019). Consistent with our results, these previous studies found that changes in the amount of (parameterized) overflow waters are associated with AMOC changes centered at 2000 m north of 60N (l.223-225 in manuscript). Hence, the relatively small transport of overflow water compared with the larger AMOC does not imply that it does not matter. A more detailed analysis of e.g., circulation and entrainment of overflow waters between the Greenland-Scotland Ridge and OSNAP-East is not the aim of this study. This is now stated in the manuscript (l.201).

We have calculated the volume transport between the Nordic Seas and the Arctic, through the Barents Sea and Fram Strait. These time series show no pronounced future changes.

5. The Nordic Seas overturning is defined from the streamfunction at 70N at an arbitrary depth such as 2000m. How does the overturning strength change when calculated in a different way (e.g., maximum streamfunction at the GSR)? How representative this latitude is in terms of the overturning changes occurring in the Nordic Seas? There is a large basin between 70N and the Greenland-Scotland-Ridge (GSR), would that contribute to the overflow change discovered at the FSC? If an arbitrary depth is deeper than where the maximum is, it reflects overturning changes in both upper limb and partial lower limb, then what does it mean physically? Since a new definition of the overturning circulation strength is used, it requires more descriptions and validations.

Our analysis is not based on a new definition of the overturning circulation and is not defined at an arbitrary depth. The Nordic Seas overturning strength in density-space (which is mainly used in this paper) is defined as the maximum value of the overturning streamfunction, following the standard

approach of e.g., Lozier et al. (2019; Science). To make this clearer, we have in the revised manuscript added the equations of the overturning strength in density- and depth-space (l.296).

As acknowledged on l.54-55 in the manuscript, “extensive water mass transformation in the Nordic Seas occurs between the Greenland-Scotland Ridge and 70N, e.g., in the Lofoten Basin (Bosse et al. 2019).” We have checked whether including this region influences the future changes in overturning by calculating the overturning for a section on the native model grid that cuts across the southern Nordic Seas. We find similar trends for the two sections (Supplementary Fig. 3). This has now been further quantified by comparing all 30-year trends between 2006-2100 for the two sections. As shown in the figure below, the correlation is high ($r=0.93$). This demonstrates that, in CESM-LE, 70N is a representative latitude in terms of overturning changes occurring in the Nordic Seas. This is now stated in the manuscript (l.303).

Figure: 30-year trends between 2006-2100 in overturning circulation for 70N and a section on the native model grid including water mass transformation (overturning) in the southeastern Nordic Seas (between GSR and 70N).

6. Why is MOC_z also presented if “overturning ... is best quantified by the meridional overturning streamfunction in density space” (lines 22-24)?

The overturning in depth-space is included in Figure 2 to highlight that you get very different projected changes depending on the method you employ (density- or depth-space). It is also included as (l.62-64) “the difference between the density-space and depth-space overturning circulation in the Nordic Seas approximates the contribution of the horizontal (gyre) circulation to MOC_σ (Zhang & Thomas 2021).”

We emphasize that MOC_z is not used in any analysis in our paper.

7. Lines 80-82: But as we know, some of the inflow of the North Atlantic join the Nordic Seas horizontal circulation and some participate in the overturning circulation. Thus, the change in the inflow may not be directly linked to the strength of the horizontal circulation.

It is true that the inflowing Atlantic water splits into an overturning circulation and an estuarine (horizontal) circulation (e.g., Eldevik & Nilsen 2013). However, this statement in the manuscript simply rules out that changes in the strength of the Atlantic inflow are directly responsible for the circulation changes in the Nordic Seas. As the Nordic Seas are highly influenced by ocean circulation variability in the North Atlantic, we believe that this is important to point out in the beginning of this paragraph.

8. Line 123: The mean overflow transport is close to observation but the subpolar overturning is larger than that observed. How does it affect the analysis that is focused on the relation between OW and overturning?

Our results and conclusions are based on numerical models which have biases and whose future projections are inherently uncertain. In the revised manuscript, we have added a new section (Discussion and implications) and expanded the discussion on model biases and uncertainties (l.237-271). The discrepancy between observed and simulated *mean* subpolar overturning strength could have many sources, and it is hard to judge whether this has a significant influence on how *variability* in subpolar overturning is affected by the Nordic Seas overturning circulation.

9. Lines 162-163: Water mass transformation in the Labrador and Irminger Seas cannot be expected to impact the transport variability in the overflow water layers.

We agree but believe that it is worthwhile to check this in the model used here (CESM-LE) and state this in the manuscript.

10. Figure 3a: Please label the x- and y-axis.

Labels have been added.

Reviewer #2 (Remarks to the Author):

Review of: "Future strengthening of the Nordic Seas overturning circulation", by Årthun et al.

In this paper, the authors analyze a strengthening of the overturning circulation in the Nordic Seas in the CESM Large Ensemble (CESM-LE) and other models from the CMIP6 archive, in the latter half of the 21st century. The authors link the circulation changes to changes in the density structure of the Norwegian Sea, which they attribute to changes in the wind stress curl over the region.

I believe there is a lot of merit in this paper, and I would love to see it published in some form. The future evolution of the Atlantic meridional overturning circulation (AMOC) is an important topic, and the role that the Nordic Seas (and Arctic in general) play in the AMOC definitely warrants more study. The finding that the overturning in the Nordic Seas may increase in the second half of the 21st century is novel and important, and may change our understanding of both the evolution of the AMOC at lower latitudes, as well as of the Arctic.

Still, there are some issues that, if addressed, would strengthen the paper. I would recommend the paper to be send back to the authors for major revisions.

We thank the Reviewer for constructive comments and suggestions. Please find below our detailed response to each of the points raised.

My primary concern is the following: even though the 'overturning circulation' in the Nordic Seas can be interpreted as horizontal circulation across sloping isopycnals, this in itself is only a kinematic statement. It does not recognize that flow through a stationary distribution of sloping isopycnals necessarily reflects a balance between advective and mixing processes. A strengthening of the horizontal circulation alone will not lead to enhanced overturning; it is only part of the story. In my mind, the question how the water mass transformations change is equally important and is not being addressed in this paper. I think an analysis of the water mass transformation processes that sustain the enhanced overturning in the second half of the 21st century would be essential to complete this study.

To assess water mass transformation processes that sustain the enhanced overturning in the second half of the 21st century, we have calculated the surface-forced water mass transformation which relates the rate of density transformation in a given density class to the surface buoyancy fluxes into that density class over its outcrop area (e.g., Marsh 2000; Petit et al. 2020). The new analysis shows that the increased water mass transformation necessary to sustain an enhanced MOC can, in part, be explained by surface forcing (new Figure 7; l.139-166). Additional water mass transformation could also be happening due to increased mixing between the boundary current and the interior, e.g., as a result of increased horizontal circulation and baroclinicity. This contribution is, however, not quantified here.

As changes in overturning circulation can be a result of enhanced horizontal circulation and/or a strengthened density gradient (isopycnal slope) across the Nordic Seas, we have also added an analysis of the horizontal density gradient across 70N (new Figure 6 and l.131-138). The new analysis shows that a weakening (strengthening) of the horizontal density gradient in the first (second) part of the 21st century was an important driver of overturning changes.

A second concern is that the analysis of the drivers of the changes in overturning was not very convincing to me. In particular the explanation that isopycnals in the western Norwegian Sea (below 1000 m nonetheless) are depressed in response to a localized change in wind stress curl and associated Ekman pumping did not convince me. It seems to me that some larger-scale adjustment of the circulation and stratification must be involved. Maybe an analysis of additional CMIP6 models, or an analysis of intra-model spread in CESM-LE, could yield some more convincing evidence for the mechanism proposed, or maybe provide additional insights. Maybe an analysis of water mass transformations might help here as well.

We have changed and extended the analysis of the potential drivers of circulation changes. The water mass transformation analysis described above suggests that surface density changes are key to increased water mass transformation toward the end of the century (as opposed to buoyancy fluxes). We find these changes in density to be strongly linked to SST changes associated with the Atlantic Multidecadal Variability (AMV) index. Horizontal circulation changes in the Nordic Seas are also strongly linked with AMV, both for the historical and future simulation in CESM-LE (new Figure 5; I.105-129). A significant link between horizontal circulation and AMV is also found for many CMIP6 models (17 models included in the analysis). We hope this new and extended analysis presents more convincing evidence for the suggested mechanism.

A third concern is that it should probably be emphasized more that this is a model analysis study. This should at least be mentioned in the abstract, if not in the title also. Also, the analysis is based mostly on a single model configuration. It is encouraging that the few available CMIP6 models support the strengthening of the overturning as found in CESM-LE (Fig. 2). But 6 models are not many, and no ensemble spread is shown in Fig. 2. What's more, Fig. 3c shows that CESM-LE is significantly higher in another metric (gyre strength) than the CMIP6 ensemble mean. One option would be to expand the CMIP6 ensemble by not just focusing on the models that provided overturning in density coordinates as standard output, but to reconstruct this overturning from (regridded) velocity and density fields (as is being done for the CESM-LE). In addition, I'm wondering if any high-resolution (HighResMIP?) simulations could be included to expand the ensemble. It would also help to also show the historical period for the CMIP6 ensemble in Fig. 2, to provide some better insights in the degree of agreement of the CMIP6 models and CESM-LE over longer time scales. Are there any observations or reanalysis products (e.g., RARE, or ASTE) that could provide some validation for the CESM-LE historical behavior? As it stands, the basis for the conclusions is rather thin.

We fully agree with the reviewer that the number of CMIP6 models used to assess future changes in overturning circulation is not many. This is also acknowledged on I.50-51. However, we remind the reviewer that we analyze more models (17) when investigating changes in the horizontal circulation (barotropic streamfunction; Figure 3). In the revised manuscript, we have added additional analysis of CMIP6 models both when discussing the relationship between AMV and horizontal circulation (Figure 5c) and when calculating the surface-forced water mass transformation (Figure 7; ensemble spread included). We hope this additional CMIP6 analysis (especially the latter) provides sufficient evidence that a future strengthening of the Nordic Seas overturning circulation is not just a feature of CESM-LE.

Historical CMIP6 simulations have been added to Figure 2. To evaluate the mean overturning strength in CESM-LE and CMIP6 we have also added four commonly used reanalysis products with a horizontal resolution of $1/4^\circ$ (OraS5, C-GLORSv7, GloSea5, GLORYS2V4). The mean overturning strength from these reanalysis products is in close agreement with CESM-LE for the period 1993-2019 (added to Figure 2; streamfunctions in Supplementary Figure 2), providing confidence in the ability of CESM-LE

to simulate water mass transformation processes in the Nordic Seas. We only consider the mean value as it is not straightforward to compare the temporal development of the CESM ensemble mean (representing only the externally-forced component of overturning change) with the reanalyses (showing a combination of external and internal variability).

Lastly, we have added a new section (Discussion and implications) where we discuss the limitations/caveats of our study, including model resolution.

Below is a list of minor and major concerns.

I. 4, 19: Nordic Seas is -> Nordic Seas are (?)

Corrected

I. 27: In my mind the statement that density-space overturning is dominated by horizontal circulation across sloping isopycnals is only a kinematic statement. It still requires mixing processes to balance advection in order to maintain sloping isopycnals. It may be my limited understanding, but to me it seems that processes like open deep convection are still critical for water mass transformations; I therefore don't see how it challenges the importance of open deep convection.

We have calculated the surface-forced water mass transformation, which relates the rate of density transformation in a given density class to the surface buoyancy fluxes into that density class over its outcrop area (e.g., Marsh 2000). This analysis shows that the reduced overturning toward the 2040s is consistent with reduced surface buoyancy fluxes and reduced convection. Considering 30-yr trends for 2006-2100, the ensemble trend correlation between MOC and MLD is, however, low ($r=0.22$) and not significant. This has been added to the text (l.171-174). In contrast, the increased overturning toward the end of the century is associated with increased surface-forced water mass transformation due to changes in surface density. Hence, deep convection is clearly important in setting intermediate densities in the Nordic Seas, but there is no evidence that *trends* in deep convection are driving the future increase in Nordic Seas overturning.

L. 37: I don't think it is emphasized enough that the conclusions are based on an analysis of numerical models. It should at least be mentioned in the abstract, if possible, in the title, and definitely in the first paragraph of the section starting at l. 37.

We now specify in the abstract and elsewhere in the text that our results are based on model simulations. We have not changed the title.

I. 44: I understand that there are only a limited number of CMIP6 models that have saved overturning in density space. Given that the main analysis is done on CESM-LE, for which the overturning is reconstructed from density and velocity fields (regridded, nonetheless), there is not really an excuse to not calculate this quantity for more models.

See response to this comment above.

II.69-70: I think here the importance of water mass transformations is neglected. How are the sloping isopycnals maintained in the face of a strengthening gyre circulation?

We have added an analysis of the horizontal density gradient across 70N (Figure 6, l.131-138) and of water mass transformation (Figure 7, l.139-166).

I. 72: How realistic is this distribution of MLD in CESM-LE, compared to observations?

In the North Atlantic, the CESM mixed layer depths (MLD) are broadly consistent with observations (Yeager and Danabasoglu 2014). Consistent with observations, the maximum MLD in the Nordic Seas is in CESM-LE located in the Greenland Sea. However, whereas the maximum MLD in observations is located in the Greenland Sea gyre (Brakstad et al 2019), it is found further east, off the coast of Svalbard in CESM. This is a common bias in climate models (Heuze 2017).

We have added a new section (Discussion and implications) where we discuss some of the model limitations and caveats (including the representation of MLDs) and how this might affect our results and conclusions.

I. 103: I'm not convinced about this argument. It's hard to imagine that the deepening of the isopycnals below 1000 m can be attributed to locally enhanced Ekman pumping.

This paragraph has been deleted. We have changed/strengthened this analysis as described above.

I like the effort to quantify the impact of enhanced Nordic Seas overturning on overturning in the subpolar North Atlantic, I think this is well done.

Thank you!

I. 173: It's not clear why the analysis is limited to only 30 ensemble members, when 40 are available?

We use the original 30 ensemble members in this study (Kay et al. 2015). This has been added to the text (l.281).

I. 188: I would be nervous calculating transport-based quantities like overturning strength based on regrided fields. The authors mention there is no compensation applied to account for Bering Strait throughflow, but is it clear that any errors introduced by regriding are negligible?

We have used regrided fields to calculate the overturning circulation across 70N as the ocean grid in CESM becomes increasingly less regular at high latitudes. We have taken great care in making sure that the regriding does not impact our results. We have calculated the transports on the native model grid (section shown in map below). The mean northward transport for regrided and original data is 7.6 Sv vs 7.7 Sv, respectively, and the standard deviation is 0.45 Sv and 0.46 Sv. The correlation (ensemble mean) is >0.9. We have also calculated the overturning on the native grid for a section cutting through the southern Norwegian Sea (Supplementary Fig. 3). Considering all 30-year trends between 2006-2100 for the two sections, the correlation is high ($r=0.93$; figure below). This has been added to the text (l.300-303).

Figure: Map showing the section (red circles) used to calculate transports on the model's native grid.

Figure: 30-year trends between 2006-2100 in overturning circulation for 70N and a section on the native model grid including water mass transformation (overturning) in the southeastern Nordic Seas (between GSR and 70N; section shown in Supplementary Figure 3).

Fig. 1: Please include lat/lon axis labels.

Added.

Fig. 2: I think it would be useful to show CMIP6 simulations for the historical period as well.

This has been added.

Fig. 3: Please include lat/lon axis labels. Maybe indicate the location of the minimum barotropic streamfunction, from which panel b is extracted. I would recommend extending the domain in panel a northward to include the entire Nordic Seas. I think it would be useful to reiterate that most panels show CESM-LE results.

We have added lat/lon axis labels and now specify in the caption that this is results from CESM-LE. We have not added the location of the minimum barotropic streamfunction as this is time dependent and will therefore be hard (messy) to display. The map extends to 80N and thus spans the northward extent of the Nordic Seas. We have moved the "(a)" in Figure 3a to make the northern part of the domain more visible.

Fig. 4: If the MOC at 70N is the subject of this paper, it is not clear to me why the authors show transects at 67N instead. The main transport features that the analysis focuses on extend only to 69N, so it's hard to see how this explains the MOC behavior further north.

This analysis and figure have been removed in the revised manuscript.

Fig. 5: Again, please reiterate that these results are from CESM-LE. Why are trends for the period 2041-2070 shown here? Why not for 2015-2100, for instance, as in Fig. 3d?

We now specify in the caption that this is results from CESM-LE. To highlight the trend reversal in the 2040s, we have chosen to show the decrease between 2011-2040 and subsequent increase between 2041-2070 in this figure.

Table 1: Modeling centers really like to be acknowledged for their efforts. One way authors can do this is by citing relevant model description papers, or by explicitly mentioning modeling centers in the acknowledgements. But most importantly, all CMIP6 model output now have citable doi's, and one can pull up references on ESGF. I know it is incredibly tedious to assemble this information, especially when the number of data sets used approaches many tens, or even hundreds. But I believe it is a responsibility we have; and publishers should support this too.

References to the various model description papers have been added to Supplementary Table 1 (if permitted we will also add these references to the CMIP6 description in the main manuscript).

-Wilbert Weijer

Reviewer #3 (Remarks to the Author):

In this paper, the authors use the Community Earth System Model Large Ensemble (CESM LE) to predict that the meridional overturning circulation in the Nordic Seas will increase in the second half of the 21st century. This contrasts with predictions that the MOC in the subpolar and subtropical North Atlantic will decrease. By analyzing the model output, the authors further attribute the increase to a strengthening of the cyclonic circulation in the Nordic Seas driven by changes in wind stress curl. Finally, they argue that these high-latitude changes may partially offset the predicted decline in overturning in the subpolar North Atlantic. The results are interesting and at a high level, would be of interest to physical oceanographers and ocean climate scientists.

Before I add my comments, I want to disclose that I'm not a subject-matter expert in numerical modeling, so I'm not in a position to evaluate the choice of model nor how this study fits within other literature that uses numerical model simulations to make inferences about past or future changes in the Atlantic meridional overturning circulation (AMOC). I'm an observational physical oceanographer involved in AMOC research who (I think) can evaluate the general methods and interpretations made in this paper and how they might be understood by non-specialists.

This perspective informs my first general comment: the authors must be more up front about (a) the fact that this study is based entirely on numerical model simulations, and (b) the pros and cons of these coarse-resolution coupled climate models for applications at high latitudes. With respect to (a), the issue starts right in the abstract with "We show that...the overturning circulation in the Nordic Seas increases..." In my opinion, this should read something like, " We show that simulated overturning circulation..." or something similar. Similarly, the first sentence in the section "Future Increase in Nordic Seas Overturning Circulation" starts out "Time series of the strength..." should be written something like "Model time series..." I realize that the focus of this study on future changes implies that some kind of model is being used, but The authors also show model output from the past, so I feel strongly that any results should be qualified and identified explicitly as being from a model, at least in the first part of the paper. As for (b), I also believe the authors should be more descriptive about the model used and its pros and cons. To their credit, they have one sentence just before section "Implications of Overturning Circulation in the Subpolar North Atlantic" noting that because of its coarse resolution, the model may over-emphasize the role of increased circulation compared to smaller-scale boundary processes, but this is not really adequate. In fact, I don't think the authors say anywhere what the spatial resolution of the model is, and how this scale compares to the natural scale of variability (the deformation radius; which is very small at these high latitudes). The authors must include this kind of basic model description if they want to publish their results in a general-audience scientific journal like Nature Communications.

Finally, the figures are very dense with information and in my opinion, some are challenging to unpack, especially for someone who is not an expert in numerical modeling. The authors have been creative in layering variables in order to minimize the number of figures, but this means one really has to spend a lot of time studying the figures and their captions to understand the message. It seems to me that the authors' target audience is the modeling community in which they are embedded. Maybe this is ok for Nature Communications-I leave this up to the editor to decide.

Otherwise, the manuscript is well-written and the figures are of generally good quality (aside from comment above about complexity, and some specific figure comments below).

We thank the Reviewer for constructive comments and suggestions. Please find below our detailed response to each of the points raised. Concerning the reviewer's general comments:

(a) more upfront about this being a model study: We agree that it is important to specify whether results are based on model output or observations. In the revised manuscript, we have made sure to specify this wherever appropriate, including in the abstract.

(b) more upfront about model caveats: We have expanded the model description, including information about horizontal resolution. The resolution of CMIP6 models has been added to Supplementary Table 1. We have also expanded the discussion about model limitations and caveats and how these could influence our results (new section Discussion and implications). We have included four commonly used ocean reanalyses products to evaluate the mean overturning strength in the models (added to Figure 2).

Specific comments:

Line 6 - This phrase demonstrates one of my primary concerns about this paper, which is that the authors do not indicate front and center that this claim ("we show that [the MOC of the Nordic Seas] increases...") is based on output from a coarse-resolution coupled climate model (see general comment above).

We now specify in the abstract and elsewhere that our results and conclusions are based on model simulations.

Line 16 - What do the authors mean by "here" in this sentence? The next sentence implies they mean the Nordic Seas and Arctic, but in fact the transformation occurs also in the North Atlantic.

Changed to "In the Nordic Seas..." (l.17)

Line 19 - "is" should be "are"

Corrected

Line 27 - Should consider referencing Petit and Koman papers here also.

We added a reference to Petit et al. (2020) and Koman et al. (2022).

Line 32 - Need to get prediction in here somewhere.

Changed to (l.33) "...the projected density-space overturning in the Nordic Seas..."

Line 37 - The authors must state the source of the time series (see general comment above). It is one thing to lean on the "Methods" section to keep the word count in the main body of the paper within limits, but I believe this is a section where the phrase "time series" should be qualified as "Model time series".

Changed to (l.42-43) "Time series of the simulated strength... in CESM-LE..."

Line 54 - The authors attribute the predicted strengthening of the MOC in density space to increase in strength of the circulation. Couldn't it also be explained by an increase in the horizontal density gradient across which the flow is oriented?

This is true. In the revised manuscript we have added additional analysis showing the future changes in the horizontal density gradient across the Nordic Seas (l.131-138; Figure 6).

Line 57 – Since the direction of circulation in the subpolar North Atlantic and Nordic Seas is cyclonic, and the two regions are connected, couldn't the pattern observed be described as a shift of the center of the joint cyclonic circulation northward?

The spatial pattern of the gyre trends in the subpolar North Atlantic is mostly aligned with the climatological pattern (Figure 3). This suggests that the gyre changes in the Nordic Seas are not simply reflecting a northward shift of the SPG-Nordic Seas circulation.

Line 63 - The authors say there is a large spread in future trends of Nordic Seas circulation, which could be contradictory to the first part of the sentence where they say all the models show an increase. This sentence needs to be made more clear.

We have removed the word “robust” from the sentence and changed to (l.76) “enhanced horizontal circulation in the Nordic Seas is a common feature across different models and forcing scenarios”

Line 70 - This addresses somewhat the question of uncertainty in whether it's increased circulation or increased horizontal density gradients.

As stated above, we have added additional analysis showing the future changes in the horizontal density gradient across the Nordic Seas. This demonstrates that both increased circulation and increased horizontal density gradients are important to future overturning changes in the Nordic Seas.

Line 84 - Are these two mechanisms independent? Wind stress curl has a direct influence on density structure through Ekman pumping/suction. The authors make this very point later. But the way this introductory sentence is phrased, it sounds like the two mechanisms are independent. This whole section needs clarification, to better differentiate between effects of wind stress, wind stress curl, changes in density structure (due to wind stress curl), etc.

This paragraph has been deleted. We have changed and extended the analysis of the potential drivers of circulation changes (new Figures 5 and 7). We hope this new and extended analysis presents more convincing evidence for the suggested mechanism.

Line 89 - I don't think "density-driven transports" is the right phrase here. It is transport changes driven by changes in the structure of the density field. Density-driven transport sounds to me like a gravity current or deep convection.

Changed to “transports driven by density changes” (l.97)

Line 93 - Is this Ekman transport?

Yes, we now specify this (l.100): “The (Ekman) transport changes...”

Line 100 - At first blush, it doesn't make sense to me that a deepening of isopycnals in the west relative to the east would lead to a stronger cyclonic circulation--quite the opposite I would think since this sounds like a relaxation of the domed isopycnals that balance the cyclonic geostrophic flow. So a

deepening of isopycnals in the west and a shoaling in the east seems like would result in a weaker northward meridional transport.

This paragraph has been deleted and replaced with a more extensive analysis of the potential drivers of circulation changes.

Line 128 - Could the increase in FSC transport also be caused by the shoaling isopycnals in the eastern Nordic Seas, which would mean higher densities at the upstream end of the channel at the sill depth?

Changes in isopycnal depth associated with wind stress curl (Ekman pumping) could contribute to density changes that impact the parameterized overflow transport in CESM-LE. The figure below, however, shows that the Ekman pumping velocities north of the Faroe-Shetland Channel (area included in overflow parameterization) increases throughout the 21st century. Hence, this is not consistent with a weakened (strengthened) overflow for the first (second) part of the 21st century. See also time series of wind stress curl added to Supplementary Figure 6.

Added to l.197-199: “The wind stress curl north of the FSC does not show trends consistent with Ekman pumping being responsible for the density changes in the Nordic Seas source waters.”

Figure: Change in Ekman pumping velocities between (right) 2031-2040 and 2006-2015, and (left) 2071-2080 and 2031-2040 in CESM-LE.

Line 193 - How small is this transport?

We have deleted “small” and added the actual value (l.293-294); “0.6 Sv averaged across ensemble members”. This net transport is related to the Bering Strait inflow.

Figure 1 - Please make the OSNAP line more visible in lower panel. Label panels a and b to match caption.

We have added labels to a,b and made the lower figure bigger to make the OSNAP line more visible.

Figure 1 - Please add latitude and longitude labels so as to better place locations of time series described in future figures.

Done

Figure 2 - is there observational evidence of decrease in MOC between 2005-2022 in Nordic Seas? This would lend confidence to the model results.

To evaluate the mean overturning strength in CESM-LE and CMIP6 models we have added four commonly used reanalyses products with a horizontal resolution of $1/4^\circ$ (OraS5, C-GLORSv7, GloSea5, GLORYS2V4). The mean overturning strength from these reanalysis products is in close agreement with CESM-LE for the period 1993-2019 (added to Figure 2), providing confidence in the ability of CESM-LE to simulate water mass transformation processes in the Nordic Seas. We only consider the mean value as it is not straightforward to compare the temporal development of the CESM ensemble mean (representing only the externally-forced component of overturning change) with the reanalyses (showing a combination of external and internal variability).

Overall, I find this revised manuscript improved. But there are several things related to my previous main comments that unfortunately haven't been addressed adequately. Some further elaboration or clarification would be helpful. Thus, I recommend the manuscript for publication after the following points are addressed.

1. Nordic Sea (NS) overturning versus the Subpolar North Atlantic (SPNA) overturning

In the manuscript (lines 226-227), it says "... the magnitude of this decline [in the overturning in the SPNA] is connected to changes in the Nordic Seas overturning circulation". The authors further explain this relationship in the response: "a stronger NS overturning corresponds to a weaker MOC decline in the SPNA (i.e., positive correlation)". This apparently refers to Figure 9c. But what Figure 9c shows is a positive relationship between the two trends that have the same units in [Sv/decade]. As such, it suggests that a weaker decline (or a stronger strengthening) in the NS overturning corresponds to a weaker decline in the SPNA overturning. What exactly is compared should be made crystal clear. Otherwise, to show how the magnitude of the trend in the SPNA overturning is connected to the NS overturning strength, one would need to change the x-axis of Figure 9c to "Nordic Sea overturning [Sv]".

2. SPNA overturning trend versus Faroe-Shetland Channel (FSC) overflow transport trend

Again, this is about the relationship between two transport trends. As shown in Figure 8, the SPNA overturning trend is positively correlated with the FSC overflow transport trend. But how the two trends with different magnitudes are connected require more elaborations. Specifically, how could a ~0.05 Sv/decade change in the FSC transport induce a ~0.2 Sv/decade overturning change in the SPNA? This would be the key to understand the implied role by the FSC overflow transport in linking the overturning changes in the NS and SPNA (lines 202-203).

The authors stated in the response that "the relatively small transport of overflow water compared with larger AMOC does not imply that it does not matter". That is true but does not seem to be relevant to my comment above on how their trends are related.

3. Overturning definition

Thanks for making it clear to me how the MOC_d is defined (i.e., the maximum of the streamfunction). But I was confused by what the authors called (line 216) "OSNAP-East overturning for each depth level" and (line 217) "OSNAP-East overturning at 2000m" (see also Figures 9c, 9d, and 9e). How is the overturning strength a function of depth? Please add more details on what it represents, and how it is related to the MOC_d .

Reviewer #2 (Remarks to the Author):

Review of: "Future strengthening of the Nordic Seas overturning circulation", first revision, by Årthun et al.

In this revision, the authors have addressed most of my concerns, and would like to thank them for their diligence in preparing their responses and revisions. There are still some loose ends, and I believe that adding a few additional –straightforward– analyses might address some of those, and hence strengthen the paper. I realize that the paper already contains a lot of information, but I hope that the authors will at least take my suggestions into consideration.

The new water mass transformation analysis has strengthened the analysis significantly. It shows contributions from both changes in buoyancy fluxes and the distribution of surface density outcrops, which is interesting. This analysis could be expanded further to address more questions about the relationship between spatial distributions of the buoyancy fluxes, isopycnal outcrops, and even the convection site), but I guess that would merit a whole new paper.

The link between MOC_sigma and the AMV is tantalizing. Without a clear connection between AMV and winds, nor with the inflow of Atlantic waters, the only other potential driver of the changes seen here seems to be warming of Atlantic waters, as alluded to in lines 134-135. I think this point should be emphasized a bit more and I think a few simple additions could clarify this issue. So rather than showing mean SST in the Nordic Seas as in Supplemental Figure 7, a measure of east-west temperature (or heat content) contrast may be more useful. One simple option is to add temperature of Atlantic inflow waters to Figure 5. But it may be more insightful to include the zonal temperature contrast (or converted to density) to Fig. 6a, and possibly show the eastern and western contributions separately. Showing similar contributions for salinity may be of interest to those readers who are wondering about the potential role of water coming out of the Arctic (see my next comment).

The paper does discuss exchanges between the Nordic Seas and the subpolar North Atlantic, but does not talk about the any exchanges with the Arctic. Admittedly, the barotropic stream function plot does not show evidence for a strong connection, but I think it would be useful to at least mention this somewhere. If the diagnosed response is a consequence of the east-west density gradient, then it would be good to reassure the reader that the dominant changes are indeed taking place in the eastern part of the basin due to warmer Atlantic inflow, while changes on the western side due to inflows from the Arctic are small.

Minor comments:

Eq. 3: I think this equation is missing the differentials.

Fig. 3a: Please clarify the units.

Fig. 4: The paper consistently talks about the Nordic Seas. This is okay in general, but sometimes the paper would benefit from adding some distinctions between the sub-basins, which are quite distinct from a hydrographic and circulation point of view. A case in point is Fig. 4, which shows a circulation feature that is limited to the Norwegian Sea. Given the limited geographic context provided by this plot to guide the reader, it may be useful to spell out that this figure only shows a limited geographical domain and that the changes are not representative for the entire Nordic Seas.

Fig. 7: Would it make sense to add these numbers of maximum buoyancy-forced overturning to Fig. 2?

Fig. 7: The aspect ratio of panel d is too large to clearly see the decline and increase of the overturning that is obvious in other plots.

ll. 139-140: I found this justification curious and interesting at the same time. As it emphasizes that a zonal density gradient should be associated with vertical shear in depth space, it raises the question why the change in west-east density gradient does /not/ project onto MOC_z. It would be

useful to share any thoughts the authors may have about this.

But to me it seems that a primary motivation of this analysis would be to explore whether surface buoyancy forcing is directly consistent with (if not, can explain) the changes in MOC_sigma; not whether it can explain the zonal density difference (which, from my perspective, is a consequence of all water mass transformation and advective processes).

I. 209: Please add 'simulated'.

-Wilbert Weijer

Reviewer #3 (Remarks to the Author):

The authors have adequately addressed my major and minor comments. I appreciate their willingness in particular to more clearly acknowledge in the manuscript that their results are based on model simulations. I recommend the article for publication.

Bergen, 28.02.2023

ANSWER TO THE REVIEWER COMMENTS

We again thank the three anonymous reviewers for constructive comments and suggestions. All issues have to the best of our knowledge been addressed in the revised manuscript. Our response to the reviews is marked using blue font. Line numbers refer to manuscript without track changes.

Best regards, on behalf of the authors,

Marius Årthun

Reviewer #1

Overall, I find this revised manuscript improved. But there are several things related to my previous main comments that unfortunately haven't been addressed adequately. Some further elaboration or clarification would be helpful. Thus, I recommend the manuscript for publication after the following points are addressed.

We thank the reviewer once again for constructive comments and suggestions. We have further modified the manuscript and hope that the issues raised by the reviewer have now been resolved.

1. Nordic Sea (NS) overturning versus the Subpolar North Atlantic (SPNA) overturning

In the manuscript (lines 226-227), it says "... the magnitude of this decline [in the overturning in the SPNA] is connected to changes in the Nordic Seas overturning circulation". The authors further explain this relationship in the response: "a stronger NS overturning corresponds to a weaker MOC decline in the SPNA (i.e., positive correlation)". This apparently refers to Figure 9c. But what Figure 9c shows is a positive relationship between the two trends that have the same units in [Sv/decade]. As such, it suggests that a weaker decline (or a stronger strengthening) in the NS overturning corresponds to a weaker decline in the SPNA overturning. What exactly is compared should be made crystal clear. Otherwise, to show how the magnitude of the trend in the SPNA overturning is connected to the NS overturning strength, one would need to change the x-axis of Figure 9c to "Nordic Sea overturning [Sv]".

The reviewer is correct that Figure 9c shows trends in overturning in the Nordic Seas and SPNA, and that the positive relationship should be interpreted as a weaker decline (or a stronger strengthening) in the Nordic Seas overturning corresponding to a weaker decline in the SPNA overturning.

We have revised the manuscript to clarify what is compared in Figure 9c,d (l.222-223): "Fig. 9c shows the relationship between trends in maximum Nordic Seas overturning and trends in OSNAP-East overturning..."

We have also modified the caption of Figure 9 to make it clearer what is compared: "(c) Correlation between trends in maximum Nordic Seas overturning strength and trends in OSNAP-East overturning..."

2. SPNA overturning trend versus Faroe-Shetland Chanel (FSC) overflow transport trend

Again, this is about the relationship between two transport trends. As shown in Figure 8, the SPNA overturning trend is positively correlated with the FSC overflow transport trend. But how the two trends with different magnitudes are connected require more elaborations. Specifically, how could a ~ 0.05 Sv/decade change in the FSC transport induce a ~ 0.2 Sv/decade overturning change in the SPNA? This would be the key to understand the implied role by the FSC overflow transport in linking the overturning changes in the NS and SPNA (lines 202-203).

The authors stated in the response that "the relatively small transport of overflow water compared with larger AMOC does not imply that it does not matter". That is true but does not seem to be relevant to my comment above on how their trends are related.

Following the reviewer's suggestion, we have in the revised manuscript elaborated on the different magnitudes of the trends in FSC overflow and SPNA overturning. It is important to note that we are not arguing that changes in FSC overflow are responsible for all the changes (trends) in SPNA overturning. This is evident, as the reviewer points out, by the weaker trends in the FSC overflow compared with overturning changes in the SPNA, and also by the correlations between NS and SPNA

overturning, which, although significant, have an explained variance of only 25% ($r \sim 0.5$). This implies that, in addition to the FSC overflow, there are also other major sources of overturning variability in the SPNA (e.g., Dickson and Brown 1994; Zou et al. 2023).

Added to I.239-243: “We emphasize that, although the ensemble trend correlation between Nordic Seas and OSNAP-East overturning is significant, a variance explained of approximately 25% ($r_{\max} \sim 0.5$; Fig. 9d) implies that other water masses than Nordic Seas overflow waters are also of major importance for subpolar overturning changes (Dickson and Brown 1994; Zou et al. 2023). Trends in FSC overflow transport (Fig.8) are also weaker than for the lower limb overturning circulation at OSNAP-East (Fig.9d).”

3. Overturning definition

Thanks for making it clear to me how the MOCd is defined (i.e., the maximum of the streamfunction). But I was confused by what the authors called (line 216) “OSNAP-East overturning for each depth level” and (line 217) “OSNAP-East overturning at 2000m” (see also Figures 9c, 9d, and 9e). How is the overturning strength a function of depth? Please add more details on what it represents, and how it is related to the MOCd.

We calculate the SPNA overturning streamfunction in density-space (MOCd), but then remap (for each time step) the overturning strength from density-space to depth-space by using the zonal mean depth of each isopycnal. We have modified the text and hope this makes it more clear how the value of the OSNAP overturning streamfunction is obtained at each depth level and how this relates to MOCd.

Added to I.215-218: “To assess whether the simulated overturning at OSNAP-East is impacted by overturning changes in the Nordic Seas we calculate the overturning streamfunction at OSNAP-East in density-space, but remap (for each year) the overturning streamfunction from density-space to depth-space by using the zonal mean depth of each isopycnal.”

To make the remapping from density to depth more transparent we have also added to Figure 9c the average density values corresponding to some depths. We’ve also changed the order of Figures 9c,d to make it clearer why we specifically show the SPNA overturning streamfunction at 2000 m depth (max correlation with NS overturning strength and depth of FSC overflow waters).

Reviewer #2 (Remarks to the Author):

Review of: "Future strengthening of the Nordic Seas overturning circulation", first revision, by Årthun et al.

In this revision, the authors have addressed most of my concerns, and would like to thank them for their diligence in preparing their responses and revisions. There are still some loose ends, and I believe that adding a few additional –straightforward– analyses might address some of those, and hence strengthen the paper. I realize that the paper already contains a lot of information, but I hope that the authors will at least take my suggestions into consideration.

We thank the reviewer once again for his thorough work and constructive comments. We have further changed the manuscript based on his comments. See our detailed responses below.

The new water mass transformation analysis has strengthened the analysis significantly. It shows contributions from both changes in buoyancy fluxes and the distribution of surface density outcrops, which is interesting. This analysis could be expanded further to address more questions about the relationship between spatial distributions of the buoyancy fluxes, isopycnal outcrops, and even the convection site), but I guess that would merit a whole new paper.

We agree that it would be interesting to further pursue the water mass transformation analysis, including, as the reviewer suggests, the spatial distribution of water mass transformation at different density levels (as in e.g., Petit et al. 2020). We further agree with the reviewer that such an analysis, however, merits a separate paper, and, hence, we have not extended this analysis here.

The link between MOC_sigma and the AMV is tantalizing. Without a clear connection between AMV and winds, nor with the inflow of Atlantic waters, the only other potential driver of the changes seen here seems to be warming of Atlantic waters, as alluded to in lines 134-135. I think this point should be emphasized a bit more and I think a few simple additions could clarify this issue. So rather than showing mean SST in the Nordic Seas as in Supplemental Figure 7, a measure of east-west temperature (or heat content) contrast may be more useful. One simple option is to add temperature of Atlantic inflow waters to Figure 5. But it may be more insightful to include the zonal temperature contrast (or converted to density) to Fig. 6a, and possibly show the eastern and western contributions separately. Showing similar contributions for salinity may be of interest to those readers who are wondering about the potential role of water coming out of the Arctic (see my next comment).

We have added a new figure (new Supp. Fig. 7; also shown below) showing the time series of density at the eastern and western side of 70N. The time series have been detrended to emphasize the multidecadal density changes relative to the linear trend seen at the eastern boundary and not at the western boundary. We also show the contribution from temperature and salinity to density changes at the eastern boundary. This figure supports the statement (l.134-137) that it is changes in temperature at the eastern boundary that are predominantly driving the changes in the zonal density gradient, although salinity becomes increasingly important toward the end of the century.

Figure: (a) Time series of simulated density changes (detrended) between 200-500 m at the western and eastern boundary of the Nordic Seas at 70N in CESM-LE. (b) Contributions from temperature (T) and salinity (S) to density changes at the eastern boundary.

The paper does discuss exchanges between the Nordic Seas and the subpolar North Atlantic, but does not talk about the any exchanges with the Arctic. Admittedly, the barotropic stream function plot does not show evidence for a strong connection, but I think it would be useful to at least mention this somewhere. If the diagnosed response is a consequence of the east-west density gradient, then it would be good to reassure the reader that the dominant changes are indeed taking place in the eastern part of the basin due to warmer Atlantic inflow, while changes on the western side due to inflows from the Arctic are small.

We hope the new analysis and figure on density changes help to shed more light on this. We have also added a sentence to the manuscript (l.134-137) emphasizing that exchanges between the western Nordic Seas and the Arctic Ocean seem to only play a minor role in the simulated density changes at 70N.

Minor comments:

Eq. 3: I think this equation is missing the differentials.

Fixed

Fig. 3a: Please clarify the units.

Added to figure caption.

Fig. 4: The paper consistently talks about the Nordic Seas. This is okay in general, but sometimes the paper would benefit from adding some distinctions between the sub-basins, which are quite distinct from a hydrographic and circulation point of view. A case in point is Fig. 4, which shows a circulation feature that is limited to the Norwegian Sea. Given the limited geographic context provided by this plot to guide the reader, it may be useful to spell out that this figure only shows a limited geographical domain and that the changes are not representative for the entire Nordic Seas.

We have modified the figure caption to make it clear that this figure only shows the eastern Nordic Seas (Norwegian Sea).

Fig. 7: Would it make sense to add these numbers of maximum buoyancy-forced overturning to Fig. 2?

We have not added these numbers to Figure 2 as we think that Figure 2 is already quite crowded.

Fig. 7: The aspect ratio of panel d is too large to clearly see the decline and increase of the overturning that is obvious in other plots.

We agree with the reviewer and have changed the format (aspect ratio) of Figure 7.

II. 139-140: I found this justification curious and interesting at the same time. As it emphasizes that a zonal density gradient should be associated with vertical shear in depth space, it raises the question why the change in west-east density gradient does /not/ project onto MOC_z. It would be useful to share any thoughts the authors may have about this.

But to me it seems that a primary motivation of this analysis would be to explore whether surface buoyancy forcing is directly consistent with (if not, can explain) the changes in MOC_{sigma}; not whether it can explain the zonal density difference (which, from my perspective, is a consequence of all water mass transformation and advective processes).

We agree with the reviewer that the main motivation for (and outcome of) the water mass transformation analysis is to assess to what extent the transformation of surface waters by air-sea buoyancy fluxes can explain future changes in the Nordic Seas overturning circulation. We have therefore moved the paragraphs on surface-forced water mass transformation to a separate section/chapter and changed the first introductory sentence (I.142-145).

The reviewer raises an interesting question about how changes in the zonal density gradient are reflected in depth-space overturning. We have, however, not investigated this further as changes in MOC_z are not the focus of our study.

I. 209: Please add 'simulated'.

Done

-Wilbert Weijer

Reviewer #3 (Remarks to the Author):

The authors have adequately addressed my major and minor comments. I appreciate their willingness in particular to more clearly acknowledge in the manuscript that their results are based on model simulations. I recommend the article for publication.

We thank the reviewer once again for their original constructive comments and suggestions that improved the manuscript.

REVIEWERS' COMMENTS

Reviewer #1 (Remarks to the Author):

I thank the authors for the detailed response and recommend the manuscript for publication. I have just two more minor points to make and hope that the authors will take them into consideration when finalizing the paper.

1. Lines 282: 'a weakening North Atlantic Ocean' reads wired. Please rephrase.
2. Figure 9: Please add to Figure 9b the time series of 'OSNAP overturning at 2000m'. That is what the Nordic Seas overturning really impacts.

Reviewer #2 (Remarks to the Author):

I'm satisfied with the responses of the reviewers, and am delighted to recommend publication of this paper in Nature Communications.

Reviewer #1 (Remarks to the Author):

I thank the authors for the detailed response and recommend the manuscript for publication. I have just two more minor points to make and hope that the authors will take them into consideration when finalizing the paper.

1. Lines 282: 'a weakening North Atlantic Ocean' reads wired. Please rephrase.

Rephrased to "a weakening North Atlantic overturning"

2. Figure 9: Please add to Figure 9b the time series of 'OSNAP overturning at 2000m'. That is what the Nordic Seas overturning really impacts.

As the data showing the trends of OSNAP overturning at 2000 m are already displayed in Figure 9d we have decided not to add an additional time series to Figure 9b.